# Unconventional self-similar Hofstadter superconductivity from repulsive interactions

Daniel Shaffer[1], Jian Wang[1] & Luiz H. Santos [1] ✉

Fractal Hofstadter bands have become widely accessible with the advent of moiré superlattices, opening the door to studies of the effect of interactions in these systems. In this work we employ a renormalization group (RG) analysis to demonstrate that the combination of repulsive interactions with the presence of a tunable manifold of Van Hove singularities provides a new mechanism for driving unconventional superconductivity in Hofstadter bands. Specifically, the number of Van Hove singularities at the Fermi energy can be controlled by varying the flux per unit cell and the electronic filling, leading to instabilities toward nodal superconductivity and chiral topological superconductivity with Chern number $\mathcal{C} = \pm 6$. The latter is characterized by a self-similar fixed trajectory of the RG flow and an emerging self-similarity symmetry of the order parameter. Our results establish Hofstadter quantum materials such as moiré heterostructures as promising platforms for realizing novel reentrant Hofstadter superconductors.

It has long been theoretically suggested that, contrary to the conventional view[1-3], superconductivity (SC) can reemerge in Landau levels in the presence of strong magnetic fields, provided there are attractive interactions[4]. More recently it has been proposed that such reentrant SC in reconstructed electron bands forming Landau levels can theoretically occur in magic angle twisted bilayer graphene (TBG)[5]. TBG and other 2D moiré superlattices are particularly attractive for realizing reentrant SC as they can host SC at zero magnetic field at low density carrier regimes[6], such that only relatively modest magnetic fields are required to achieve the quantum limit of Landau levels. However, several challenges have stood in the way of observing reentrant SC in experiment, among them the role of repulsive interactions that make quantum Hall states natural competitors of such reentrant SC in Landau levels.

In this work we propose that this issue can be circumvented in Hofstadter bands that, unlike Landau levels, have a finite bandwidth $W$[7,8], allowing a weak-coupling renormalization group (RG) treatment of repulsive electronic interactions. This is especially relevant for moiré systems since their nanometer scale unit cells enable the realization of Hofstadter bands in experimentally accessible magnetic fields at which the magnetic flux per super unit cell $\Phi = B A_{\mathrm{uc}}$ is comparable to the flux quantum $\Phi_0 = 2\pi\hbar/e = 2\pi$ in natural units[9-15]. Beyond the rich phenomena in Hofstadter-Chern insulators[16-30], a recent classification[31] has shown that Hofstadter bands may support novel Hofstadter superconductors (HSC) characterized by spontaneous breaking of the magnetic translation group (MTG) symmetries[32-34], leading to multi-component finite momentum Cooper pairing similar to pair-density wave states[35]. HSCs embody a new form of reentrant superconductivity in Hofstadter bands, in which the large flux per unit cell makes the magnetic length comparable to the lattice scale, thus generalizing the Landau level reentrant SC state.

Although pairing in Hofstadter bands has been studied earlier using mean-field calculations with phenomenological attractive interactions[36-42], no microscopic mechanism leading to this attraction has so far been proposed. Here we show that HSCs can arise from repulsive interactions due to the competition of electronic orders near Van Hove singularities (VHS) that provide a logarithmic enhancement of the density of states (DOS)[43]. Such a scenario of competing orders near VHSs underlies several proposed mechanisms of unconventional superconductivity through repulsive interactions, for example in cuprates[44-46], doped graphene[47-50], and moiré graphene superlattices[51-59]. Furthermore, we go beyond mean field by using an

[1]Department of Physics, Emory University, 400 Dowman Drive, Atlanta, GA 30322, USA. ✉e-mail: luiz.santos@emory.edu

RG analysis[60–62], extending it to the new realm of Hofstadter electronic bands and uncovering a new pathway to realize reentrant superconductivity in moiré superlattices. This approach allows us to treat the interplay of all logarithmically divergent instabilities on equal footing, and thus to additionally study the competition of superconductivity with charge/spin density wave (CDW/SDW) orders, thus going beyond earlier mean-field studies of CDW and SDW in Hofstadter systems in refs. 63 and 64,65, respectively. The RG analysis also provides an alternative scenario to fractionalization in Hofstadter bands[66–75].

Famously, the Hofstadter spectrum has a fractal nature characterized by a self-similar structure as a consequence of the MTG symmetries[8,76]. We find that, remarkably, some aspects of this self-similarity are passed on to the RG flow and some of the resulting instabilities. First, we find that for all flux values, the RG flow has a particular fixed trajectory that is equivalent to multiple copies of the RG flow in the absence of the magnetic flux. We therefore refer to it as a self-similar fixed trajectory. As there are in principle many other fixed trajectories of the RG flow, it is not a given that the self-similar trajectory is reached for given set of interactions. Nevertheless we find that the self-similar trajectory is reached in our model in some cases. Second, in those cases the superconducting instability occurs by the same VHS mechanism as proposed in cuprates[44–46], but the resulting order parameter also repeats in a self-similar fashion in the magnetic Brillouin zone. The self-similarity of the order parameter can be expressed as an emergent symmetry, which we call the self-similar symmetry, and which we show implies a highly non-local character of the order. These self-similarity properties illustrate how the MTG symmetries of the Hofstadter system can lead to novel phenomena via the VHS patch RG mechanism.

As a proof of principle, we work with the repulsive fermionic square-lattice Hofstadter-Hubbard (HH) model with on-site interaction $U > 0$ and flux $\Phi = 2\pi p/q$ that is a rational multiple of the flux quantum. Importantly, we focus on the regime $q \sim 1$ in which the Hofstadter bands have a bandwidth $W$ comparable to that of the original band at zero field, which allows us to investigate electronic instabilities in a controlled weak-coupling regime $U/W \ll 1$. While a hexagonal lattice would better approximate twisted bilayer graphene, which is the best studied superconducting moiré system, we establish our results on the square lattice since it still allows us to capture the essential correlation effects in Hofstadter bands while working with a simpler band structure, as shown in section Hofstadter-Hubbard VHS patch model and interactions. Nevertheless, we stress that while the competition of electronic orders and their resulting instabilities can depend on the underlying lattice and interactions, the weak-coupling RG framework developed here is of general applicability, and thus represents an important step towards the investigation of electronic instabilities in a wider class of two dimensional Hofstadter superlattices, including moiré graphene.

Additionally, the square HH model can more easily be realized in cold atom systems[77–86], although the focus in that field has been on bosonic[87–92] and time-reversal invariant fermionic[93–98] HH models (note that the latter coincides with the regular fermionic Hofstadter-Hubbard model at $q = 2$, i.e., at $\pi$-flux). In addition, more recently single layer cuprates exhibiting critical temperatures close to their bulk values have been fabricated[99], opening an avenue for realizing twisted cuprate moiré systems with square lattices for which our model may be directly applicable. Such twisted heterostructures have recently been studied theoretically[100–104], with few-layer twisted interfaces already realized in experiment[105,106]. It remains to be seen whether Hofstadter physics can be realized in twisted cuprates, but, if it is, a reentrant HSC phase may be possible in this system.

The MTG symmetries play a key role in our analysis. In particular, they imply the presence of $2q$ VHSs per Hofstadter band, as shown in Fig. 1. The magnetic flux $\Phi = 2\pi p/q$ thus acts as a knob controlling the number of VHSs in the system, which completely alters the RG flow and thus the possible instabilities of the system. This is well illustrated by

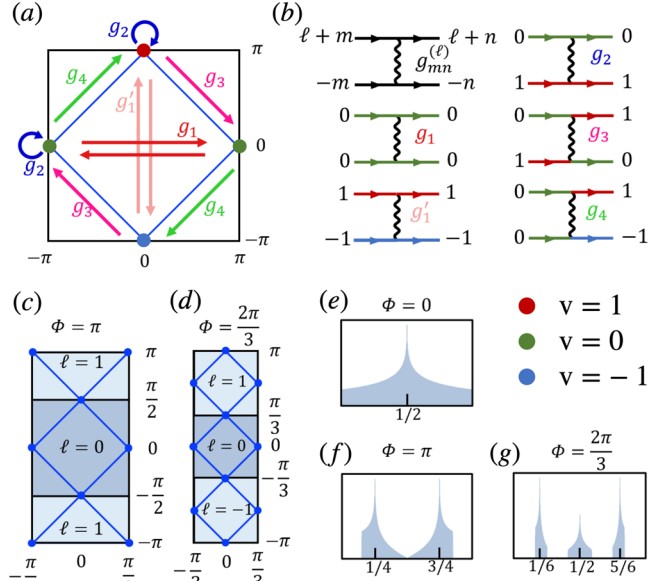

**Fig. 1 | Van Hove singularities and relevant interactions in the square Hofstadter model.** VHSs are shown at **a** zero, **c** $\pi$, and **d** $2\pi/3$-flux, and **e–g** the corresponding peaks in the density of states at indicated fillings. Due to the MTG symmetry, the magnetic Brillouin zone (MBZ) splits into $q$ (energy degenerate) reduced magnetic Brillouin zones (rMBZ) labeled with $\ell = 0, ..., q-1$. In each band there are a total of $2q$ VHSs occurring at momenta $\mathbf{K}_{\ell,\mathrm{v}} = \left((1+\mathrm{v})\frac{\pi}{q}, (\mathrm{v}+2p\ell)\frac{\pi}{q}\right)$, such that there is a pair of VHSs in each rMBZ labeled with a VHS index $\mathrm{v} = 0, \pm 1$, with the identification of VHS $\ell, 1$ and $\ell + 1, -1$. Arrows in **a** and the Feynman diagrams in **b** show the types of interaction processes considered in the RG analysis: intra-VHS processes $g_1$ and $g_{1'}$ (red and light red); inter-VHS forward scattering $g_2$ (blue); exchange $g_3$ (magenta); and pair-hopping $g_4$. The VHS index color-coded in **a** and **b** as red/green/blue for $\mathrm{v} = 1, 0, -1$, respectively, and the black diagram shows the additional rMBZ indices $\ell, m, n = 0, ..., q-1$ carried by the coupling constants $g_{mn}^{(\ell)}$, $\ell$ denoting the total momentum of the interacting pair.

the two distinct reentrant HSC phases that we find at $\pi$-flux (i.e., $q = 2$) and at $2\pi/3$-flux ($q = 3$). For the former case, we identify a nodal SC phase that respects all MTG symmetries as the winning RG instability at 1/4 and 3/4 lattice filling, even with perfect nesting in the competing SDW channel that is degenerate with the SC channel in the absence of the magnetic field[107]. For $q = 3$, we find that, at 1/6 and 5/6 lattice filling, SC and SDW are nearly degenerate when both are at perfect nesting, while SDW is strongly favored at half-filling. A small symmetry-allowed detuning from perfect nesting in the SDW therefore favors the pairing instability at 1/6 and 5/6 filling, which necessarily breaks a subset of the MTG symmetries[31]. We find that the resulting SC state is a fully gapped chiral topological phase with Chern number $\mathcal{C} = \pm 6$ that preserves a $\mathbb{Z}_3$ subgroup of the MTG. Surprisingly, this phase realized for $q = 3$ also possesses an emergent self-similarity symmetry due to the RG flow approaching a special self-similar fixed trajectory that exists as another consequence of the MTG symmetries. We identify this self-similar fixed trajectory for all $q$, implying that long-range self-similar HSC states can be competing instabilities at flux values beyond those studied numerically in this work.

## Results

### Hofstadter-Hubbard VHS patch model and interactions

We consider the nearest-neighbor square-lattice repulsive HH Hamiltonian

$$H = -\sum_{\langle \mathbf{r}\mathbf{r}'\rangle\sigma} t\, e^{2\pi i A_{\mathbf{r}\mathbf{r}'}} c_{\mathbf{r}\sigma}^{\dagger} c_{\mathbf{r}'\sigma} + h.c. - \mu \sum_{\mathbf{r}\sigma} c_{\mathbf{r}\sigma}^{\dagger} c_{\mathbf{r}\sigma} + $$
$$+ U \sum_{\mathbf{r}} n_{\mathbf{r}\uparrow} n_{\mathbf{r}\downarrow} = H_0 + H_{\mathrm{int}},$$

(1)

with $U > 0$ where $\mu$ is the chemical potential, $n_{\mathbf{r}\sigma}$ is the number operator with spin $\sigma = \uparrow, \downarrow$ at site $\mathbf{r} = (x, y) \in \mathbb{Z}^2$, and $A_{\mathbf{rr}'} = \int_{\mathbf{r}}^{\mathbf{r}'} \mathbf{A} \cdot d\mathbf{r}/\Phi_0 = \frac{p}{q}x(1 - \delta_{yy'})$ corresponding to a flux per unit cell $\Phi = 2\pi p/q$ that is a rational multiple of the flux quantum $\Phi_0$. We work in the Landau gauge with vector potential $\mathbf{A} = xB\hat{y}$ and set the lattice constant $a = 1$. Note that while time-reversal symmetry (TRS) is broken by the orbital effect, we neglect the Zeeman splitting in our analysis and retain the full SU(2) spin rotation symmetry, implying that our weak-coupling analysis is applicable in the regime $E_Z \ll \Delta \ll W$, where $E_Z$ is the Zeeman splitting and $\Delta$ is the characteristic energy scale of the electron instabilities. The interesting regime of spin polarized Hofstadter bands case merits a separate discussion which is outside the scope of this work.

In addition to TRS, the vector potential breaks the translation symmetry $T_x$ along the $x$ direction. However, the magnetic translation $\hat{T}_x = T_x e^{2\pi i aBy/\Phi_0}$ remains a symmetry of the Hamiltonian. $\hat{T}_x$ and the unbroken translation $T_y = \hat{T}_y$ along the $y$ direction generate the non-Abelian magnetic translation group (MTG) satisfying $\hat{T}_x \hat{T}_y = \omega_q^p \hat{T}_y \hat{T}_x$ with $\omega_q = e^{2\pi i/q}$ being the $q$th root of unity. Point group symmetries of the original Hamiltonian in the absence of the magnetic field similarly give rise to their magnetic versions with appropriate gauge transformations of the vector potential. For example, this includes inversion symmetry that guarantees a logarithmic pairing instability, as well as the original $C_4$ symmetry that becomes $\hat{C}_4 = C_4 e^{-2\pi i xyB/\Phi_0}$, where the additional gauge transformation rotates $\mathbf{A} = xB\hat{y} \to yB\hat{x}$. The $\hat{C}_4$ symmetry will play a role below when we consider the instabilities of the $\pi$-flux Hofstadter Hamiltonian.

The commutation relations imply that $\hat{T}_x^q$ and $\hat{T}_y$ commute with each other and the Hamiltonian, effectively enlarging the unit cell along the $x$ direction. We correspondingly define operators $c_{\mathbf{R},s,\sigma} = c_{s\hat{\mathbf{x}} + \mathbf{R},\sigma}$ with $s = 0, ..., q-1$ being the sublattice index defined modulo $q$ and $\mathbf{R} = (qj, y)$ with $j, y \in \mathbb{Z}$ labeling the extended unit lattice cites. Bloch's theorem then applies to these operators and we can write the Hofstadter Hamiltonian $H_0$ in momentum space using $c_{\mathbf{k}s\sigma} = \frac{1}{\sqrt{N}} \sum_{\mathbf{R}} e^{-i\mathbf{k} \cdot (s\hat{\mathbf{x}} + \mathbf{R})} c_{s\hat{\mathbf{x}} + \mathbf{R},\sigma}$, with $N$ being the total number of unit cells and where the quasi-momentum $\mathbf{k}$ is defined on the folded magnetic Brillouin zone (MBZ) $\mathbf{k} = (k_x, k_y) \in [-\pi/q, \pi/q] \times [-\pi, \pi]$. In this basis

$$H_0 = -\sum_{\mathbf{k}s} (2t \cos(k_y + sQ) + \mu) c_{\mathbf{k}s\sigma}^\dagger c_{\mathbf{k}s\sigma} - \\ - \sum_{\mathbf{k}\langle ss'\rangle} t e^{-ik_x(s-s')} c_{\mathbf{k}s\sigma}^\dagger c_{\mathbf{k}s'\sigma} , \quad (2)$$

and the magnetic translation symmetries act as $\hat{T}_x c_{\mathbf{k}s\sigma} \hat{T}_x^\dagger = e^{-ik_x} c_{\mathbf{k}+\mathbf{Q},s+1,\sigma}$ and $\hat{T}_y c_{\mathbf{k}s\sigma} \hat{T}_y^\dagger = e^{-ik_y} c_{\mathbf{k}s\sigma}$, with $\mathbf{Q} = \frac{2\pi p}{q}\hat{y}$.

The Hofstadter Hamiltonian $H_0$ can then be diagonalized as $H_0 = \sum_{\mathbf{k}\alpha\sigma} \varepsilon_\alpha(\mathbf{k}) d_{\mathbf{k}\alpha\sigma}^\dagger d_{\mathbf{k}\alpha\sigma}$ using a unitary transformation

$$d_{\mathbf{k}\alpha\sigma} = \sum_s \mathcal{U}_\alpha^s(\mathbf{k}) c_{\mathbf{k}s} . \quad (3)$$

Note that there is a large freedom in choosing the U(1) phases in $\mathcal{U}_\alpha^s(\mathbf{k})$. For concreteness, we take $\mathcal{U}_\alpha^{s+1}(\mathbf{k} + \mathbf{Q}) = \mathcal{U}_\alpha^s(\mathbf{k})$, which endures a canonical transformation under MTG for the band operators: $\hat{T}_x d_{\mathbf{k}\alpha\sigma} \hat{T}_x^\dagger = e^{-ik_x} d_{\mathbf{k}+\mathbf{Q},\alpha,\sigma}$ and $\hat{T}_y d_{\mathbf{k}\alpha\sigma} \hat{T}_y^\dagger = e^{-ik_y} d_{\mathbf{k}\alpha\sigma}$. Furthermore, we fix the remaining gauge freedom by taking $\mathcal{U}_\alpha^1(\mathbf{k}) \in \mathbb{R}$. This choice makes it clear that the $\hat{T}_x$ symmetry implies a $q$-fold degeneracy of each band, $\varepsilon_\alpha(\mathbf{k}) = \varepsilon_\alpha(\mathbf{k} + \mathbf{Q})$. This means we can further restrict the quasi-momentum to a reduced magnetic Brillouin zone (rMBZ) $\mathbf{p} = (p_x, p_y) \in [-\pi/q, \pi/q]^2$ and define $d_{\mathbf{p}\ell\alpha\sigma} = d_{\mathbf{p}+\ell\mathbf{Q},\alpha\sigma}$ where $\ell = 0, ..., q-1$ is the magnetic patch index defined modulo $q$ as defined in ref. 31 (see Fig. 1c, d). We also refer to $\ell$ as the rMBZ magnetic flavor index to distinguish it from the VHS indices introduced below.

Unlike earlier mean-field analyses of the fermionic HH model[36–39,41,63–65] (see also ref. 108 who studied SC in the related

Aubry-André model), here we investigate the instabilities driven by repulsive on-site interactions due to diverging DOS at the VHSs. In the square-lattice Hofstadter model, the VHSs occur at electron fillings that are odd multiples of $1/(2q)$ (counting spin), i.e., in half-filled Hofstadter bands. In each band there are a total of $2q$ VHSs occurring at momenta $\mathbf{K}_{\ell,v} = \left((1+v)\frac{\pi}{q}, v\frac{\pi}{q}\right) + \ell\mathbf{Q}$ which we label with the VHS index $v = 0, 1$[109]. Note that the VHSs thus lie at the images of the original VHSs of the square lattice at zero flux under a rescaling of the momentum by $1/q$, which is a consequence of the self-similarity property of the Hofstadter spectrum[16,76] that also implies that the Fermi surfaces are composed of $q$ touching squares for any Hofstadter band (see Fig. 1a, c, d).

Within this weak-coupling framework we can project the interactions onto the Fermi surfaces formed by a single band $\alpha$, neglecting all other bands and expand the dispersions around patches centered at the VHS momenta $\mathbf{K}_{\ell,v}$, obtaining a VHS patch model that we will analyze in section RG analysis using fermionic RG[44,45,47,62,110]. We thus define the patch model operators $d_{\mathbf{p}\ell v\alpha\sigma} = d_{\mathbf{p}+\mathbf{K}_{\ell,v},\alpha,\sigma}$ with $\mathbf{p}$ a small momentum expanded around a patch centered at $\mathbf{K}_{\ell,v}$. For bookkeeping purposes, we include a redundancy in our description and allow $v = -1$ with the identification $\mathbf{K}_{\ell,-1} \equiv \mathbf{K}_{\ell-1,1}$ which makes the VHS and magnetic flavor indices conserved quantities in Feynman diagrams we use in the RG analysis.

We then project $H_{int}$ in Eq. (1) onto the patches obtaining an effective interaction Hamiltonian

$$H_{int} \to H_{int}^{(\alpha)} = \frac{1}{2} \sum_{\substack{\ell mn \\ uvw,\sigma\sigma'}} g_{m,v;n,w}^{(\alpha;\ell,u)} d_{\ell+n,u+w,\alpha,\sigma}^\dagger d_{-n,-w,\alpha,\sigma'}^\dagger d_{-m,-v,\alpha,\sigma'} d_{\ell+m,u+v,\alpha,\sigma} , \quad (4)$$

where $\ell, m, n = 0, ...q-1$ are magnetic flavor indices, u, v, w = 0, ±1 are the VHS indices, and

$$g_{m,v;n,w}^{(\alpha;\ell,u)} = U \sum_s \mathcal{U}_\alpha^s(\mathbf{K}_{\ell+n,u+w}) \mathcal{U}_\alpha^s(\mathbf{K}_{-n,-w}) \mathcal{U}_\alpha^{s*}(\mathbf{K}_{-m,-v}) \mathcal{U}_\alpha^{s*}(\mathbf{K}_{\ell+m,v}) \quad (5)$$

are the coupling constants corresponding to interactions between electrons with total momenta $u(\pi, \pi)/q + \ell\mathbf{Q}$, dressed by form factors originating from the unitary transformation Eq. (3). Henceforth we will consider a fixed band $\alpha$ and drop the index where it is clear from context.

As there are $2q$ VHSs, the number of coupling constants grows quickly with $q$, which manifests the MTG action in momentum space. Taking hermiticity, MTG symmetries, and redundancy of the VHS indices into account, there are a total of $\mathcal{O}(q^2)$ independent coupling constants that can be classified into five processes according to their VHS indices:

$$g_{mn}^{(\ell)1} = g_{m,0;n,0}^{(\ell,0)} \qquad g_{mn}^{(\ell)1'} = g_{m,1;n,1}^{(\ell,0)} \\ g_{mn}^{(\ell)2} = g_{m,0;n,0}^{(\ell,1)} \qquad g_{mn}^{(\ell)3} = g_{m,0;n,-1}^{(\ell,1)} \quad (6) \\ g_{mn}^{(\ell)4} = g_{m,0;n,1}^{(\ell,0)} ,$$

as shown in Fig. 1b. $g_1$ and $g_{1'}$ correspond to intra-patch processes for $v = 0, \pm 1$ VHSs, respectively, $g_2$ ($g_3$) is an inter-patch process without (with) exchange, and $g_4$ is a pair-hopping process. Note that in the absence of TRS, not all coupling constants are necessarily real. In addition to relations imposed by hermiticity, the coupling constants also satisfy $g_{mn}^{(\ell)j} = g_{m-1,n-1}^{(\ell+2),j}$ as a consequence of the MTG action on the fermion operators $\hat{T}_x d_{\mathbf{p}\ell v\sigma} \hat{T}_x^\dagger = e^{-ip_x} d_{\mathbf{p},\ell+1,v\sigma}$. In particular, for odd $q$ all coupling constants can be expressed in terms of $g_{mn}^{(0)j}$. For even $q$, all coupling constants can be expressed in terms of either $g_{mn}^{(0)j}$ or $g_{mn}^{(1)j}$, with an additional relation $g_{mn}^{(\ell)j} = g_{m-q/2,n-q/2}^{(\ell)j}$; see the Supplementary Material for further relations satisfied by the coupling constants. By virtue of the MTG symmetries the coupling constants Eq. (6) thus organize into processes that resemble those in zero magnetic field.

As we will see in section RG analysis, this has the important implication that the RG equations exhibit a degree of self-similarity that we will elucidate below.

## RG analysis

In this section we extend the RG analysis developed previously for the half-filled square lattice[44,45,107] and the quarter-filled hexagonal lattice[47] with 2 and 3 VHSs, respectively, to the patch model with $2q$ VHSs presented above. The competing instability channels fall into two classes: particle-particle channels with momentum transfers $\ell\mathbf{Q}$; and particle-hole channels with momentum transfers $(\pi, \pi)/q + \ell\mathbf{Q}$. Due to the MTG symmetries, all the susceptibilities are independent of the magnetic flavor indices $\ell$, and the two relevant susceptibilities are $\Pi_{pp}(\ell\mathbf{Q}) \approx \nu_0 \ln^2\Lambda/T$ and $\Pi_{ph}((\pi, \pi)/q + \ell\mathbf{Q}) \approx d_{ph}\nu_0 \ln^2\Lambda/T$ where $\Lambda$ is the high energy cutoff, $T$ is the temperature and $\nu_0 \ln\Lambda/E$ is the DOS at energy $E$ above the VHS[47,56,107]. Here we introduce the standard phenomenological detuning parameter $d_{ph} = \Pi_{ph}/\Pi_{pp} \in [0, 1]$ to account for possibly imperfect nesting in the particle-hole channels due to additional symmetry-allowed terms that break particle-hole symmetry at half-filling or for chemical potentials slightly away from the VHSs[107].

Performing the one-loop RG procedure (see Supplementary Material) and keeping only the most diverging $\ln^2$ corrections, we obtain the flow equations for the coupling constants. The full expressions are given in the Supplementary Material and can be represented symbolically as:

$$
\begin{aligned}
\dot{g}_{mn}^{(\ell)1} &= -g_{mk}^{(\ell)1}g_{kn}^{(\ell)1} - g_{mk}^{(\ell)4}g_{nk}^{(\ell)4*} \\
\dot{g}_{mn}^{(\ell)1'} &= -g_{mk}^{(\ell)1'}g_{kn}^{(\ell)1'} - g_{km}^{(\ell)4*}g_{kn}^{(\ell)4} \\
\dot{g}_{mn}^{(\ell)2} &= d_{ph}\left(g_{mk}^{(\ell+n-k)2}g_{kn}^{(\ell+m-k)2} + g_{mk}^{(\ell+n-k)4*}g_{kn}^{(\ell+m-k)4}\right) \\
\dot{g}_{mn}^{(\ell)3} &= 2d_{ph}g_{-n-k,-m-k}^{(\ell+m+n+k)3}\left(g_{m,-n-k}^{(k)2} - g_{mn}^{(k)3}\right) + d_{ph}g_{-n-k,-m-k}^{(\ell+m+n+k)4}\left(g_{n,-m-k}^{(k)4*} - 2g_{n,m-1}^{(k)4*}\right) \\
&\quad + d_{ph}g_{-n-k,-n-\ell-1}^{(\ell+m+n+k)4}g_{n,m-1}^{(k)4*} \\
\dot{g}_{mn}^{(\ell)4} &= -g_{mk}^{(\ell)4}g_{kn}^{(\ell)1} - g_{mk}^{(\ell)4}g_{kn}^{(\ell)1'} + d_{ph}\left(g_{k-\ell-m-n,-\ell-m}^{(\ell+n-k)2}g_{kn}^{(\ell+m-k)4} + g_{mk}^{(\ell+n-k)4}g_{k+1,n+1}^{(\ell+m-k-1)2}\right) \\
&\quad + d_{ph}g_{-n-k,-m-k}^{(\ell+m+n+k)4}\left(g_{-m-k+1,n+1}^{(k)2} - 2g_{1-k,-m,-k-n}^{(k-1)3}\right) + d_{ph}g_{1-k,-m,-k-n}^{(\ell+m+n+k)4}g_{1-k,-m,-k-n}^{(k-1)3} + \\
&\quad + d_{ph}g_{-n-k,-m-k}^{(\ell+m+n+k)3}g_{-m-k,n}^{(k)4} + d_{ph}\left(g_{-n-k,-n-\ell}^{(\ell+m+n+k),2} - 2g_{-n-k,-m-k}^{(\ell+m+n+k)3}\right)g_{mn}^{(k)4} .
\end{aligned}
$$

$$(7)$$

The dot denotes the derivative with respect to the running RG time $t = \Pi_{pp}(E) = \nu_0 \ln^2\Lambda/E$, with high energy modes integrated above the energy scale $E$. For $q = 1$, i.e., zero flux, Eq.(7) reduces to the standard result for the half-filled square lattice in ref. [44] (in this case $g_{1'} = g_1$ by $C_4$ symmetry that is otherwise broken at non-zero flux). Recall that in that case repulsive Hubbard interactions lead to degenerate d-wave SC and SDW orders, with the degeneracy being lifted either by imperfect nesting or subleading terms in RG[44].

For $q \neq 1$ the RG equations (7) in principle allow for a large number of fixed trajectories that characterize the instabilities of the Hofstadter metal. Despite the apparent complexity of these equations, by grouping the coupling constants into the $g_{mn}^{(\ell)1}, g_{mn}^{(\ell)1'}, g_{mn}^{(\ell)2}, g_{mn}^{(\ell)3}, g_{mn}^{(\ell)4}$ processes according to the VHS patch index structure, we see that the form of these equations is similar to the RG equations in the absence of the magnetic flux, i.e., for $q = 1$. In particular, it can be verified that they admit a fixed point trajectory characterized by $g_{mn}^{(\ell)j} = g_j/\sqrt{q}$, i.e., coupling constants independent of the magnetic flavor indices and depending only on the VHS patch indices. Plugging this ansatz into the RG equations (7), one can directly verify that $g_j$ satisfy the same set of equations as for $q = 1$. As a consequence, the resulting instability and its properties such as critical exponents are identical to those in the $q = 1$ system, and we thus refer to such solutions as self-similar fixed trajectories. We note that this property extends to all classes of Hofstadter systems beyond the one studied here, given that the MTG symmetries are preserved and provided the weak-coupling regime is valid.

Though we show that the self-similar solutions exist, at the beginning of the RG flow local interactions produce bare couplings $g_{mn}^{(\ell)j}(t = 0)$ that in general have a dependence on the magnetic flavor.

It is therefore not a given that the self-similar trajectory is reached by the RG flow, and we find for example that it is not reached with repulsive Hubbard interactions for $q = 2$. We do find, on the other hand, that with the same repulsive Hubbard interactions in top and bottom Hofstadter bands for $q = 3$, the coupling constants do tend asymptotically to this self-similar fixed trajectory. The existence of such nontrivial self-similar behavior in the RG equations and their relation to unconventional SC is one of the main results of this work.

## Vertices and susceptibilities

Under the RG flow some of the coupling constants diverge at some finite RG time $t_c$, indicating an instability of the Fermi surface (see Fig. 2a and c). To study these instabilities, we introduce the following test vertices and study their flow:

$$
\begin{aligned}
H_{SC} &= \Delta_{m;\nu}^{(\ell)} i\sigma_{\sigma\sigma'}^y d_{\ell+m,\nu,\sigma}^\dagger d_{-m,-\nu,\sigma'}^\dagger + h.c. \\
H_{CDW} &= \rho_{m;\nu}^{[\ell]} d_{\ell+m,-\nu,\sigma}^\dagger d_{m,1+\nu,\sigma} \\
H_{SDW} &= \mathbf{M}_{m;\nu}^{[\ell]} \cdot \boldsymbol{\sigma}_{\sigma\sigma'} d_{\ell+m,-\nu,\sigma}^\dagger d_{m,1+\nu,\sigma'}
\end{aligned}
$$

$$(8)$$

with summation over the indices implied. $\Delta_{m;\nu}^{(\ell)}$, $\rho_{m;\nu}^{[\ell]}$, and $\mathbf{M}_{m;\nu}^{[\ell]}$ are the SC, CDW, and SDW order parameters respectively with momentum transfers $\ell\mathbf{Q}$ for SC and $(\pi, \pi)/q + \ell\mathbf{Q}$ for the density waves. Note that by hermiticity, $\rho_{m;0}^{[\ell]} = \rho_{m+\ell;1}^{[1-\ell]*}$, and similarly $M_{m;0}^{[\ell]} = M_{m+\ell;1}^{[1-\ell]*}$, which therefore belong to the same channel in the RG flow. As shown in the Supplementary Material, the CDW and SDW flow equations decouple into $q^2$ channels each: $\tilde{\rho}_{k;\nu}^{[\ell]} = \sum_m \omega_q^{mk} \rho_{m;\nu}^{[\ell]}$ for CDW and similarly for SDW. This is consistent with the fact that being charge-0 instabilities, CDW and SDW transform as 1D irreducible representations (irreps) of the MTG. Similar CDW orders have been found numerically in a real-space mean-field analysis of a spinless fermionic HH model on a hexagonal lattice, although their MTG irreps have not been established[63]. Note also that we do not find the previously proposed $(\pi, \pi)$ SDW as a potential instability[64,65].

As shown in ref. [31], unlike the charge-0 orders, the charge-2 SC orders transform according to $q$ or $q/2$ irreps of the MTG for odd and even $q$, respectively. This means that in general there are $q$ or $q/2$ degenerate flows in the SC channels corresponding to each choice of $\ell$ in Eq. (8), with even and odd $\ell$ being non-degenerate for even $q$. In addition, when $q$ is even the SC order parameter decouples into $\Delta_{m;\nu}^{(\ell,\pm)} = \Delta_{m;\nu}^{(\ell)} \pm \Delta_{m+q/2;\nu}^{(\ell)}$, with $\Delta_{m;\nu}^{(\ell,+)}$ (even under $\hat{T}_x^{q/2}$) and $\Delta_{m;\nu}^{(\ell,-)}$ (odd under $\hat{T}_x^{q/2}$) flowing independently, consistent with the fact that there are four $q/2$-dimensional irreps in this case[31].

The vertex RG flow equation are shown schematically in Fig. 3. Observe that the coupling constants $g_{mn}^{(\ell)1}$ and $g_{mn}^{(\ell)1'}$ only contribute to the flow of the SC vertices, $g_{mn}^{(\ell)3}$ only contributes to the CDW flow, while $g_{mn}^{(\ell)2}$ contributes to the flow of both CDW and SDW vertices. $g_{mn}^{(\ell)4}$, on the other hand, contributes to all the channels, with a similar structure in the flow of the coupling constants themselves in Eq. (7). SC is thus generally favored by negative $g_{mn}^{(\ell)1}$ and $g_{mn}^{(\ell)1'}$. Importantly, although these are positive initially in the repulsive Hubbard model, they can potentially change sign due to the $|g_4|^2$ term in their flow in Eq. (7). We indeed find this to be the case for $q = 2$ and $q = 3$, as shown in Fig. 2a and c.

In order to establish which instability actually takes place, we additionally consider the flow of the susceptibilities $\chi_I$ where $I = \Delta_{m;\nu}^{(\ell)}, \tilde{\rho}_{k;\nu}^{[\ell]}, \tilde{M}_{k;\nu}^{[\ell]}$ corresponding to the instability. The susceptibilities flow as $\dot{\chi}_I = d_I|I(t)/I(0)|^2$ with $d_\Delta = 1$ and else $d_I = d_{ph}$[56,62]. The leading instability corresponds to $\chi_I$ that diverges most strongly at $t_c$, around which they generally diverge as $\chi_I(t) \propto (t_c - t)^{1-2\alpha_I}$ with some critical exponent $\alpha_I$ that needs to be larger than $1/2$ for the instability to occur. A representative flow for $q = 2$ ($q = 3$) is shown in Fig. 2b, d. In that case we find that SC is the leading instability with critical exponent $\alpha_{SC} \approx 0.73$ ($\alpha_{SC} \approx 0.65$). The exponent is computed as the final value of

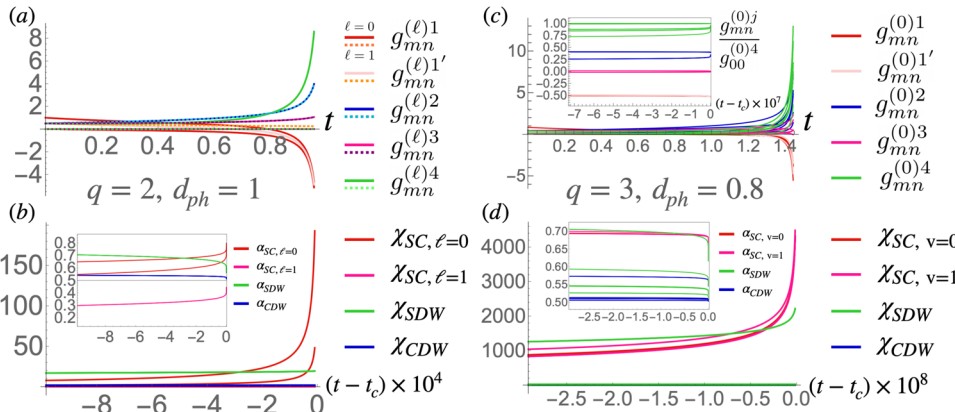

**Fig. 2 | RG flow of coupling constants and susceptibilities.** The flow of the coupling constants $g_{mn}^{(\ell)j}$ with $\ell = 0, 1$ (solid and dashed lines, respectively), $j = 1, 1', 2, 3, 4$ (red, light red, blue, magenta, and green, respectively), and $m, n = 0, ..., q-1$ are shown for **a** $q = 2$ at 1/4 filling at perfect nesting $d_{ph} = 1$; and for **c** $q = 3$ at 1/6 filling with $d_{ph} = 0.8$ ($U = 1$ in arbitrary units in all plots). The instability occurs at $t_c = 0.98$ and $t_c = 1.46$ for $q = 2$ and 3, respectively. The flows for $q = 2$ and 3 are otherwise qualitatively similar, and both are similar to the flow in the absence of the magnetic field: note that while all coupling constants are initially positive or vanishing, $g_{mn}^{(0)1}$ and $g_{mn}^{(0)1'}$ eventually change sign, leading to effective attraction in the pairing channel. The inset in **c** shows the $q = 3$ flow normalized by $g_{00}^{(0)4}$ which shows that the self-similar fixed trajectory $g_{mn}^{(\ell)j} = g_j/\sqrt{q}$ is reached at the end of the flow, as indicated by curves of the same color approaching the same value (we also find

$g_1 = g_{1'}$). **b** $q = 2$ RG flow of the susceptibilities $\chi_I$ with $I$ corresponding to SC with Cooper pairs with zero momentum ($\ell = 0$, red) or momentum $\mathbf{Q} = \frac{2\pi p}{q}\hat{\mathbf{y}}$ ($\ell = 1$, magenta), SDW (green) or CDW (blue). Initially $\chi_{SDW}$ is the fastest growing susceptibility, but eventually The $\ell = 0$ SC susceptibility takes over. The inset shows the corresponding critical exponents $\alpha_I = \left(1 - \log_{t_c - t}\chi_I\right)/2$ for the same range of RG times $t$. The largest exponent at the end of the flow is $\alpha_{SC,\ell=0}(t_c) \approx 0.73$. (**d**) Shows that analogous plots for $q = 3$, but in this case the $\ell = 0$ and 1 SC channels are degenerate so only the former is plotted; in this case red and magenta colors indicate the suscpetibilities at v = 0 and 1 VHS points, respectively, which contribute to the same SC channel. The largest exponent at the end of the flow is $\alpha_{SC}(t_c) \approx 0.65$. Color online.

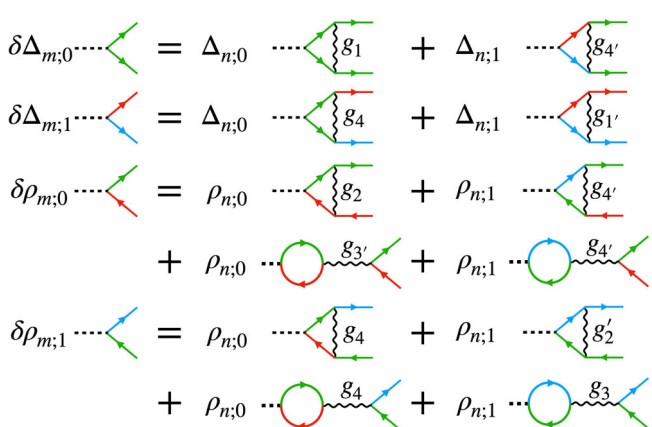

**Fig. 3 | The 1 loop Feynman diagrams contributing to the SC and CDW vertex corrections.** The VHS index structure is shown (green for v = 0, red/blue for v = ±1, respectively). SDW Diagrams for SDW vertex corrections are the same as the CDW diagrams with $M$ instead of $\rho$, except for the loop diagrams in the second lines that vanish for SDW vertices. The propagators additionally carry the magnetic flavor indices, not shown in the figure (these can be found in the Supplementary Material).

$\alpha_I(t) = \left(1 - \log_{t_c - t}\chi_I\right)/2$, with a representative plot of $\alpha_I(t)$ shown in the inserts of Fig. 2b and d.

## Resulting instabilities

We studied the RG equations for $p/q = 1/2, 1/3$ and 2/3, with the results summarized in Table 1. For zero flux, $q = 1$ and we recover the results for the square lattice with repulsive interactions at half-filling[44,45,107]. In that case it is found that a d-wave SC and SDW are degenerate within one loop at perfect nesting in the SDW channel, $d_{ph} = 1$, with d-wave SC winning for any $d_{ph} < 1$. This has been interpreted as SDW fluctuations leading to an effective attraction in the d-wave SC channel. In the flow of the coupling constants this is reflected in the initial growth of $g_4$ that pushes $g_1$ to become negative and eventually diverge. We observe a qualitatively similar

RG flow for $q = 2$ and 3 as seen in Fig. 2a and c. We now analyze the resulting instabilities for those cases.

Unlike the $q = 1$ case, for $q = 2$ we find that an SC instability occurs already at perfect nesting in both Hofstadter bands (with critical exponent $\alpha_{SC} \approx 0.77 > 0.5$). As shown in ref. 31, in this case the SC orders belong to one of four one dimensional irreducible representations (irreps) of the MTG determined by the gap function being even or odd under $\hat{T}_y$ and $\hat{T}_x$. The SC phase that wins in our RG calculation is even under both $\hat{T}_y$ and $\hat{T}_x$, which corresponds to $\Delta_{m;v}^{(1)} = 0$ and $\Delta_{0;v}^{(0)} = \Delta_{1;v}^{(0)}$ respectively. Furthermore, we find that $\Delta_{m;0}^{(0)} = -\Delta_{m;1}^{(0)}$ (see Fig. 4a), which implies that the gap function is odd under the magnetic $\hat{C}_4$ rotation. We note that this is an exceptional case, as for $q > 2$ the gap function necessarily breaks one of the MTG symmetries, and must either break the $\hat{C}_4$ symmetry or break the remaining MTG symmetry[31]. Only when the gap function is both even or both odd under $\hat{T}_x$ and $\hat{T}_y$, as in the present case, can it also have a well-defined $\hat{C}_4$ symmetry.

The RG analysis only determines the gap function at the VHSs, so our approach does not determine the gap function $\Delta_m^{(0)}(\mathbf{p})$ along the entire Fermi surface (with $\Delta_{m;v}^{(0)} = \Delta_m^{(0)}(\mathbf{K}_{0,v})$). In principle, this issue can be addressed by using a method that extends the RG calculation to the entire Fermi surface, for example a function RG calculation or a two-step RG approach combined with a random phase approximation type calculation (see e.g., ref. 111); however, this is an involved computation that is beyond the scope of this work. For $q = 2$ we can circumvent this issue by using the fact that a $\hat{T}_x$ and $\hat{T}_y$ symmetric gap function odd under $\hat{C}_4$, which we refer to as a d-wave gap function, has a unique nearest-neighbor form in the $c_{\mathbf{k}s}$ basis, namely:

$$\Delta_{ss'}^{(d)}(\mathbf{k}) = \Delta_0\left(\cos k_x \sigma_{ss'}^x - \cos k_y \sigma_{ss'}^z\right) \tag{9}$$

The anti-symmetry of this order parameter under $\hat{C}_4$ symmetry can be checked directly by using

$$\hat{C}_4 c_{\mathbf{p}+\ell\mathbf{Q},s\sigma}\hat{C}_4^\dagger = \frac{1}{q}\sum_{s'\ell'}\omega_q^{-p(ss'+\ell s'+\ell's)}c_{\hat{\mathbf{p}}+\ell'\mathbf{Q},s'\sigma} \tag{10}$$

**Table 1 | Summary of instabilities $I = \Delta$, $\tilde{M}_k^{[\ell]}$, and $\tilde{\rho}_k^{[\ell]}$, (SC, SDW, and CDW, respectively) found in the RG analysis for $q = 2$ (column two) and $q = 3$ at ($d_{ph} = 1$, next three columns) and away from ($d_{ph} = 0.8$, last three columns) perfect nesting in the particle-hole channels**

| | $q = 2$, $d_{ph} = 1$ | $q = 3$, $d_{ph} = 1$ | | | $q = 3$, $d_{ph} = 0.8$ | | |
|---|---|---|---|---|---|---|---|
| | 1/4, 3/4 | 1/6 | 1/2 | 5/6 | 1/6 | 1/2 | 5/6 |
| $I$ | $\Delta^{(d)}$ | $\tilde{M}_0^{[0]}$, $\tilde{M}_0^{[1]}$ | $\tilde{M}_2^{[0]}$, $\tilde{M}_1^{[1]}$ | $\tilde{M}_0^{[0]}$, $\tilde{M}_0^{[1]}$ | $\Delta$ | $\tilde{M}_2^{[0]}$, $\tilde{M}_1^{[1]}$ | $\Delta$ |
| $\alpha_I$ | 0.77 | 0.68 | 0.71 | 0.68 | 0.65 | 0.65 | 0.65 |
| Symmetries | $\hat{T}_x$, $\hat{T}_y$, $\hat{C}_4(-1)$ | $\hat{T}_x(\omega_3^{-1/2})$ | $\hat{T}_x(\omega_3^{-1})$ | $\hat{T}_x(\omega_3^{-1/2})$ | $\hat{T}_x\hat{T}_y(\omega_3^n)$, $\hat{S}$ | $\hat{T}_x(\omega_3^{-1})$ | $\hat{T}_x\hat{T}_y(\omega_3^n)$, $\hat{S}$ |
| | | $\hat{T}_y(\omega_3^{-1/2})$ | $\hat{T}_y(\omega_3^{3/2})$ | $\hat{T}_y(\omega_3^{-1/2})$ | | $\hat{T}_y(\omega_3^{3/2})$ | |

For $q = 3$ the subcolumns indicate the filling corresponding to the VHSs at which the instabilities are found (for $q = 2$ the same instability occurs at both 1/4 and 3/4 VHS fillings). Second row indicates the critical exponent $\alpha_I$ of the corresponding instability and the last row shows its symmetry; values in parentheses indicate the phase picked up by the order parameter under the symmetry, e.g., $\Delta^{(d)} \xrightarrow{\hat{C}_4} -\Delta^{(d)}$. Recall that $\omega_q = e^{2\pi i/q}$.

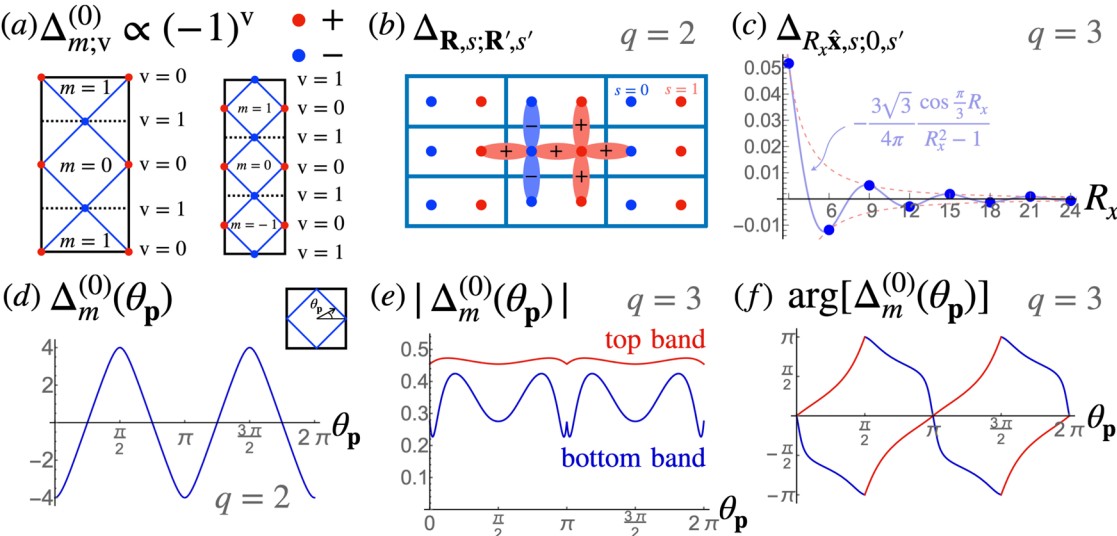

**Fig. 4 | Properties of the gap functions. a** Gap functions at the VHS obtained from the RG analysis for $q = 2$ at perfect nesting (left) and for $q = 3$ at $d_{ph} = 0.8$ in the top and bottom Hofstadter bands (right). In both cases the gap function changes sign between the two VHSs $v = 0, 1$. Here we focus on pairing with zero total momentum $\ell = 0$, with pairings for $\ell \neq 0$ determined by MTG symmetries. **b** Real-space structure of the gap function for $q = 2$ even under $\hat{T}_1$ and $\hat{T}_2$ and odd under $\hat{C}_4$, shown within a single magnetic unit cell (the pattern repeats in all cells). **c** Profile of the gap function $\Delta_{R_x\hat{x},s;0,s'}$ for $q = 3$ as a function of the horizontal magnetic unit cell separation $R_x$ between Cooper pairs (with lattice constant $a = 1$). Note that the gap function oscillates between each unit cell and decays as $1/R_x^2$ at long distances. See the Supplementary Material for more details. **d** The projection onto the Fermi surface of the gap function for $q = 2$ shown in **b** as a function of the angle $\theta_\mathbf{p}$ along the Fermi surface within the rMBZ (note that $\Delta_m^{(\ell)}$ are equal within each patch $m$). Note that the gap crosses zero, indicating nodes in the fermionic spectrum. **e, f** The projection onto the Fermi surface of the model gap function for $q = 3$ for the top (red) and bottom (blue) bands that agrees with the gap function found in the RG analysis (color online). Note that the magnitude of the gap function never vanishes as shown in **e**, implying that the fermionic spectrum is fully gapped (the sharp features at $\theta_\mathbf{p} = 0, \pi$ are due to the corners of the Fermi surface). The phase of the projected gap functions, however, winds by $\pm 4\pi$ around the Fermi surface in the top and bottom bands respectively, as shown in **f**, implying each $\Delta_m^{(\ell)}$ contributes $\pm 2$ to the Chern number. Plots (**c**–**e**) are given in arbitrary units as the magnitude of the gap function is not determined within the weak-coupling theory.

where $\bar{\mathbf{p}} = (-p_y, p_x)$ (one can also check that the RHS in Eq. (10) is an eigenstate of $\hat{T}_1$). Figure 4b shows the corresponding gap function $\Delta_{\mathbf{R},s;\mathbf{R}',s'}$ in real-space in the $c_{\mathbf{R}s}$ basis. The gap function $\Delta_0^{(0)}(\mathbf{p})$ is then obtained by projecting $\Delta_{ss'}^{(d)}(\mathbf{k})$ onto the band basis $d_{\mathbf{k}\alpha}$ (see Supplementary Material for details). Importantly, the resulting gap is nodal (see Fig. 4d).

A gap function of this form has been considered as a toy model of a nodal $d$-wave superconductor in a magnetic field in ref. 112, but without a microscopic justification or a consideration of its symmetries presented here (indeed, the gap function in that model does not transform as a proper irreducible representation of the MTG for $q > 2$). The $\pi$-flux superconductor on a square lattice has also previously been studied using quantum Monte Carlo at half-filling, i.e., at the Dirac nodes of the normal spectrum, where a so-called $ds$ SC phase has been found[113]. The corresponding gap function, which we simply refer to as $s$-wave, has the form $\Delta_{ss'}^{(s)}(\mathbf{k}) = \Delta_0 \left( \cos k_x \sigma_{ss'}^x + \cos k_y \sigma_{ss'}^z \right)$ and we find

that it is precisely the $\hat{T}_x$, $\hat{T}_y$ symmetric gap that is even under $\hat{C}_4$, and therefore distinct from the phase we find in RG at VHS fillings.

For $q = 3$, the $\tilde{M}_2^{[0]}$ and $\tilde{M}_1^{[1]}$ SDW susceptibilities (degenerate by hermiticity) diverge first in the middle band, while SC and $\tilde{M}_0^{[0]}$ (or the degenerate $\tilde{M}_0^{[1]}$, again by hermiticity) SDW diverge first in the top and bottom bands. This suggests strongly competing instabilities in the top and bottom bands that likely remain degenerate at perfect nesting as in the $q = 1$ case, and a small detuning from perfect nesting generally favors SC instabilities. We find that for $d_{ph} = 0.8$, SC is a clear winner in the top and bottom bands at 1/6 and 5/6 fillings, but SDW remains the apparent leading instability at half-filling. Remarkably, we find that when SC is the winning instability, the RG flow approaches the self-similar fixed trajectory $g_{mn}^{(\ell)j} = g_j/\sqrt{q}$ within numerical accuracy, as shown in the inset in Fig. 2c. We therefore expect the results for the $q = 1$ case to generalize in this case. Observe that this is unlike the $q = 2$ case for which the self-similar fixed trajectory is not reached.

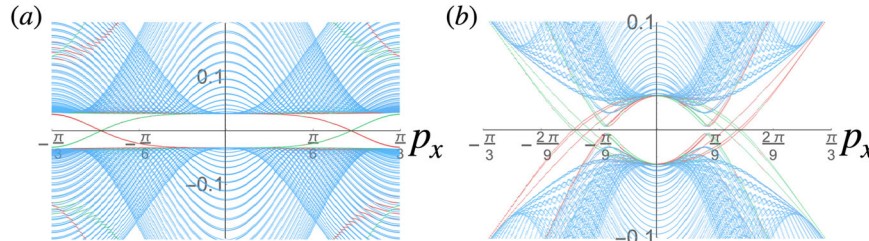

**Fig. 5 | Edge modes in the BdG spectrum of the Hofstadter SC for $q = 3$.**
Cylindrical boundary conditions open in the $y$ direction were taken for the self-similar $\hat{T}_x\hat{T}_y$ symmetric gap function Eq. (12) at **a** 5/6 and **b** 1/6 filling (chemical potential $\mu = \pm 2.44$, respectively, with $t = 1$ and $\Delta_0 = 0.02$ in **a** and 0.2 in **b**, taking 100 extended unit cells along the $y$ direction; see Methods for details of the BdG Hamiltonian and Supplementary Material for more details of the calculation). The spectra are colored according to a weighted inverse participation ratio with green and red indicating states localized to the top and bottom edges of the cylinder,

respectively, while blue indicates bulk states. In **a** there are pairs of crossing edge modes at zero energy around $p_x = \pm 2\pi/9$, and we find that each is three-fold degenerate, corresponding to Chern number 6. In **b** there are six right-moving and six left-moving zero energy edge modes are located around $p_x = \pm \pi/6$, giving a total Chern number $-6$. Observe that the edge modes of the same color move in opposite direction in **a** and **b**. Localized edge modes at higher energies that do not cross zero energy are the normal state edge modes that connect to higher energy Hofstadter bands not shown in the figure.

Indeed, the SC phase we find in the top and bottom bands satisfies $\Delta_{m;\nu}^{(0)} = \Delta_{m+1;\nu}^{(0)}$ and $\Delta_{m;0}^{(0)} = -\Delta_{m;1}^{(0)}$, similar to the $q = 2$ and $q = 1$ cases. Unlike those cases, however, there is no natural interpretation of these relations in terms of MTG and $\hat{C}_4$ symmetries. As shown in ref. 31, in this case the gap function transforms according to a 3D irrep of the MTG and necessarily breaks at least one of $\hat{T}_x$ or $\hat{T}_y$, and any $\hat{C}_4$ symmetric gap breaks all of the MTG symmetries. In order to determine the symmetries of the resulting degenerate ground states it is necessary to include fourth order terms in the Ginzburg-Landau free energy, which goes beyond the 1 loop RG analysis. Computing the fourth order term using an approximation scheme outlined in the Supplementary Material, we find that there are three degenerate ground states symmetric under $\omega_q^m \hat{T}_x \hat{T}_y$ with $m = 0, 1, 2$, and $\hat{C}_4$ is therefore broken. This determines the rest of the $\Delta_{m;\nu}^{(0)}$ order parameters for $\ell \neq 0$, so below we will focus on the form of $\Delta_{m;\nu}^{(0)}$ only.

The $\Delta_{m;\nu}^{(0)} = \Delta_{m+1;\nu}^{(0)}$ condition extended to $\Delta_m^{(0)}(\mathbf{p}) = \Delta_{m+1}^{(0)}(\mathbf{p})$ on the full rMBZ implies an additional symmetry that emerges under the RG flow, which we refer to as a self-similarity symmetry $\hat{S}$. This symmetry acts on the gap function as

$$\Delta(\mathbf{p}) \xrightarrow{\hat{S}} \hat{T}_x(\mathbf{p})\Delta(\mathbf{p})\hat{T}_x(-\mathbf{p}) \tag{11}$$

(in contrast to the canonical action of $\hat{T}_x$ itself, which acts as $\Delta(\mathbf{p}) \xrightarrow{\hat{T}_x} \hat{T}_x(\mathbf{p})\Delta(\mathbf{p})\hat{T}_x^T(-\mathbf{p})$[31]). Stated another way, $\hat{S}$ acts as $\hat{T}_x$ on the particle sector but as $\hat{T}_x^T$ on the hole sector in the Nambu space of the Bogoliubov-de Gennes (BdG) formalism. We refer to this symmetry as self-similarity because in momentum space it implies that the gap function is independent of the magnetic flavor index and thus repeats three times (or $q$ times generalized to other $q$).

Though the self-similarity symmetry $\hat{S}$ acts in a simple way in momentum space, its action on the gap function in the sublattice basis $c_{\mathbf{k}s}$ is not trivial and it takes $\Delta_{ss'}^{(\ell)}(\mathbf{k}) \xrightarrow{\hat{S}} \Delta_{s-1,s'+1}^{(\ell)}(\mathbf{k} + \mathbf{Q})$. In the real-space basis $c_{\mathbf{R}s}$, the action of this symmetry has a highly non-local character: $\Delta_{\mathbf{R}s;\mathbf{R}'s'} \xrightarrow{\hat{S}} e^{-i\mathbf{Q}\cdot(\mathbf{R}-\mathbf{R}')}\sum_{X\in q\mathbb{Z}} \mathrm{sinc}\left[\frac{\pi}{q}(X+2)\right] \Delta_{\mathbf{R},s+1;\mathbf{R}'+X\hat{\mathbf{x}},s'-1}$, where $\mathrm{sinc}(x) = \sin x / x$ (see Supplementary Material for details of the change of basis transformation). In particular, if $\Delta_{\mathbf{R}s;\mathbf{R}'s'}$ is symmetric under $\hat{S}$, it decays as $1/(R_x - R_x')^2$, implying a long-range order and an obstruction to constructing fully localized Wannier states of the BdG Hamiltonian (see Fig. 4c).

As for $q = 2$, our method does not determine the form of the gap function along the whole Fermi surface, and either a chiral or a nodal form of the gap within the rMBZ matches the $\Delta_{m;0}^{(0)} = -\Delta_{m;1}^{(0)}$ relation. In this case symmetry does not completely fix the form of the gap function, but we find that the simplest form of the extended gap function respecting the $\hat{S}$ symmetry and matching the RG result at

VHSs can be obtained in the sublattice basis:

$$\Delta_{ss'}^{(0)}(\mathbf{k}) = \Delta_0\left[1 - \cos k_x - \cos(k_y - (s-s')Q)\right] \tag{12}$$

Though as mentioned above this gap function cannot be written down in real space using nearest-neighbor terms, it can be constructed using an extended $s$-wave gap function $\Delta_{\mathbf{rr}'}^{(S)} = \Delta_0(\delta_{\mathbf{rr}'} - \sum_{\mathbf{a}}\delta_{\mathbf{r},\mathbf{r}'+\mathbf{a}}/2)$ where $\mathbf{a}$ is summed over all nearest neighbors of the square lattice. The real-space order parameter can then be obtained by repeatedly applying the $\hat{S}$ symmetry, $\Delta_{\mathbf{R}s;\mathbf{R}'s'} = \sum_j \hat{S}^j\left[\Delta_{\mathbf{rr}'}^{(S)}\right]$. We then obtain the extension $\Delta_m^{(\ell)}(\mathbf{p})$ by projecting onto the band basis $d_{\mathbf{p}\ell\alpha}$ (see Supplementary Material), and find that the resulting order parameter is fully gapped and chiral, $\Delta_m^{(0)}(\mathbf{p}) \sim e^{\pm 2i\theta}$ with $\pm$ for the upper and lower bands respectively, contributing a Chern number of $\pm 2$ (see Fig. 4e–f). An important consequence of the $\hat{S}$ symmetry is the three-fold degeneracy of the BdG spectrum of the fermionic excitations, which therefore implies that the total Chern number of this phase is $\pm 6$. We verify this numerically for the $\hat{T}_x\hat{T}_y$ symmetric gap function by computing the BdG spectrum with cylindrical boundary conditions periodic in the $x$ direction and open in the $y$ direction (taking advantage of the gap function being short-ranged in the latter). The resulting spectrum is shown in Fig. 5.

## Discussion

To summarize, we have investigated the nature of electronic instabilities on the square-lattice Hofstadter-Hubbard model using a weak-coupling renormalization group analysis to characterize competing electronic orders when the Fermi level is brought near a manifold of $2q$ VHSs and the flux per unit cell is $\Phi = 2\pi p/q$. The RG analysis allows for the treatment of competing instabilities on equal footing, revealing how the progressive elimination of high energy modes renormalizes the bare repulsive interactions and opens low energy instability channels. One of the main results of our analysis is the demonstration of the existence of self-similar fixed trajectories of the RG flow related to the RG equations at zero field. Remarkably, we find that the self-similar fixed trajectory is reached by the RG flow for $q = 3$ (but not for $q = 2$) when the SC instability occurs. The existence of a self-similar structure in the RG flow of Hofstadter systems is a novel result that illustrates the power of the magnetic translation group in constraining the low energy instabilities.

We analyzed the RG equations for two representative cases, with the results summarized in Table 1. First, for $p/q = 1/2$ corresponding to the TR-symmetric $\pi$-flux phase we have identified nodal d-wave superconducting instabilities near 1/4 and 3/4 fillings. The nodal order parameters are odd under the magnetic rotation $\hat{C}_4$ and have unusual real-space structure (see Fig. 4b) giving rise to a gapless spectrum of Bogoliubov quasiparticles that manifest themselves in a

linear-in-temperature specific heat. Importantly, unlike the zero flux case, the SC instability is leading even when the nesting is perfect in the density wave channels. Second, for $p/q = 1/3, 2/3$ corresponding to $\pm 2\pi/3$-flux lattices our analysis uncovers the existence of a novel chiral topological superconductors near 1/6 and 5/6 fillings. These TRS broken paired states break $\hat{C}_4$ symmetry while preserving a $\mathbb{Z}_3$ subgroup of the MTG, thus realizing a $\mathbb{Z}_3$ Hofstadter superconductor classified in ref. 31. Having a gapped bulk spectrum, these novel phases are characterized by a bulk Chern number topological invariant $\mathcal{C} = \pm 6$, which accounts for a chiral phase with 6 net chiral Majorana edge modes. A universal experimental signature of such phases is a quantized thermal Hall coefficient $\kappa_{xy}/T = 6 \times (\pi^2 k_B^2/3h)$. Even more remarkably, the chiral phases occur when the system flows to the self-similar trajectory of the RG equations and as a result possess a self-similarity symmetry $\hat{S}$ defined in Eq. (11) that forces the real-space order parameters to be long-ranged, providing another experimental signature of these phases. Moreover, since the self-similar trajectory is present for all $q$, the self-similar HSC instability is viable for all values of the magnetic flux. The prediction of unconventional nodal and self-similar topological superconductivity in partially filled Hofstadter bands from intrinsic electronic interactions are the two main results of this work.

In addition, we found several closely competing spin density wave instabilities that break MTG symmetries and that may be of experimental interest in their own right. Below the transition temperature, these states can coexist with the HSC states and can give rise to rich and complex phase diagram similar to those of high $T_c$ superconductors[114]. Moreover, the multi-component nature of the HSC order parameters implies that vestigial density wave orders may appear in the vicinity of the SC instability and can provide an experimental signature of these phases[115].

Recently, Hofstadter systems have experienced a renaissance caused by the advent of 2D moiré superlattices realizing large magnetic fluxes in laboratory accessible magnetic fields. For nearly four decades, Hofstadter bands have been predominantly studied as platforms for the quantum Hall effect, following the seminal work of Thouless and collaborators[116] that showed that it is a consequence of the topology of filled Hofstadter bands. However, the connection between Hofstadter systems and the quantum Hall effect is but one aspect of the physics embodied by fractal electronic bands. This work invites a broader view on the potentialities of Hofstadter quantum materials. Rather surprisingly, our RG analysis predicts that superconductivity can be driven by repulsive interactions in Hofstadter systems, surprising not only just because of the role played by electronic interactions, but also because it implies the formation of Cooper pairs in large magnetic fields that cause a strong orbital effect commonly viewed as detrimental for superconductivity. Our analysis therefore establishes a new microscopic mechanism for the realization of reentrant superconductivity in Hofstadter materials, which could be within near-term experimental reach in moiré superlattices.

In particular, our theoretical findings on the square-lattice Hofstadter-Hubbard model may directly inform the realization of reentrant Hofstadter superconductivity in a number of experimental platforms, including optical lattices[78,81–83,86] and twisted cuprate moiré systems[105,106]. Moreover, the RG framework developed here for the square lattice can be directly generalized to other Hofstadter systems. A particularly interesting direction is to extend this formalism to effective lattice models describing the band structure of magic angle twisted bilayer graphene where $2\pi/3$ and $\pi$ flux lattices can be realized at accessible magnetic fields $B \sim 8$ T and $B \sim 12$ T, respectively. Similar fields would be required away from the magic angle in which case the bands are not as flat and our weak-coupling analysis may apply more directly. In that regard, the experimental observation[117] of reentrant behavior in twisted bilayer graphene and other moiré systems with small Zeeman splitting ($\lesssim 2$ meV) may offer a promising route to search for emergence of Hofstadter superconductivity, enabled by the competition of electronic orders in the complex manifold of Van Hove singularities present in moiré Hofstadter superlattices.

The RG theory can also be extended to the case of spin polarized bands for materials in which the Zeeman splitting is strong. In that regime triplet Hofstadter superconductivity may become possible. Recent observations of triplet SC in twisted trilayer graphene[118,119] as well as Bernal stacked bilayer graphene[120] indicate that this may be another promising route to realizing HSCs. Of course in all these systems, including TBG, strong correlation effects may play an important role, which are known to affect the Hofstadter spectrum[75,121,122] and have been seen to lead to fractional and ferromagnetic states in experiment in the Hofstadter regime[13,14,21]. Recently, Hofstadter superconductivity has also been studied in the strong coupling limit using a mean-field theory[123]. Including these strong coupling effects in the RG framework is likely necessary to properly study Hofstadter superconductivity in magic angle TBG due to the presence of flat bands. This is a challenging task we leave for a future study, but we expect a nontrivial interplay of HSC with these strongly correlated states that can give rise to even more unconventional phases.

## Methods

In this work we extended the standard parquet RG analysis of VHS patch models previously used to study SC from repulsive interactions on square and hexagonal lattices[44,45,47,56,58,62,107] to the HH model with half-filled Hofstadter bands. The details of this calculation are presented in the accompanying Supplementary Material. In this analysis we introduce test vertices corresponding to all possible instabilities of the Fermi surface and study their RG flow. The resulting flows are shown in Fig. 2. The chief advantage of this method is that it allows us to go beyond mean field and study all possible instabilities on equal footing, letting the system decide which instability wins. Throughout the RG analysis we make extensive use of the MTG symmetries to identify different channels of the RG flow.

Since the RG calculation only determines the order parameter at the VHS points, it is necessary to extend it in some way to determine the nature of the resulting phase (chiral or nodal). In principle, one needs to extend the RG calculation to the whole BZ, which is computationally prohibitive already for moderate $q$. Even solving the self-consistent gap equation for a constant Hubbard interaction numerically is quite challenging. We therefore adopt a simpler approach and construct an ansatz gap function in real space in the $c_{r\sigma}$ basis first (e.g., standard $s$- or $d$-wave gap functions with up to nearest-neighbor terms, etc.) consistent with the symmetries of the ground state, and then projecting onto the Hofstadter band of interest via $d_{k\alpha\sigma} = \sum_s \mathcal{U}_a^s(\mathbf{k}) c_{\mathbf{k}s\alpha}$ with the band index $\alpha$ fixed. The details of this projection are presented in the Supplementary Material and the resulting gap function extensions are presented in Fig. 4.

With the gap function extension we can then study the Bogoliubov-De Gennes (BdG) spectrum of the fermionic excitations of the system, including its topological properties. The BdG spectrum is also needed to derive Ginzburg-Landau free energy. As shown in ref. 31, due to the HSC order parameter belonging to a multidimensional irreducible representation of the MTG for $q > 2$, it is necessary to expand the free energy up to fourth order in powers of the order parameter (the one-loop approximation in the RG being equivalent to a second order approximation of the free energy). We use this in order to establish the MTG symmetry of the $q = 3$ HSC phases, as outlined in the Supplementary Material. The BdG Hamiltonian with the gap function extension expressed in real space also allows us to study the system on a cylinder (i.e., with periodic boundary conditions along the $x$ direction but open in the $y$ direction) in order to identify the topological edge modes. The resulting edge modes are shown in Fig. 5, with the details of the calculation presented in the Supplementary Material.

## Data availability
Data sharing not applicable to this article as no datasets were generated or analyzed during the current study.

## Code availability
All numerical codes in this paper are publicly accessible at[124].

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

## Acknowledgements

We thank Claudio Chamon, Dmitri Chichinadze, Ben Feldman, Gil Refael and Jörg Schmalian for helpful discussions. L.H.S. acknowledges support from the U.S. Department of Energy, Office of Science, Basic Energy Sciences, under Award DE-SC0023327, and from startup funds at Emory University.

## Author contributions

L.H.S. conceived and designed the project. D.S. carried out the RG calculations. D.S and J.W. performed the numerical analysis of the RG instabilities and analyzed the properties of the paired states. L.H.S supervised the project. All authors contributed to the writing of the paper.

## Competing interests

The authors declare no competing interests.
