## [Peer Review File · Nature Communications]

Reviewers' Comments:

Reviewer #1:

Remarks to the Author:

The article titled "Unconventional Self-similar...Repulsive interactions" by Shaffer et. Al. addresses the problem of finding the ground-state in an interacting system under strong magnetic field. While in the free electron system, strong magnetic fields lead to Quantum Hall (QH) and fractional QH effects and possible SC within, the authors address a similar problem in a lattice (they choose to focus on square), where the Landau levels are replaced by Hofstadter bands: the reconstructed electronic levels in strong fields. The latter introduced one key modification that is absent in free-electron systems: the Van-Hove singularities (VHS) at appropriate fillings. The authors focus on these specific fillings and address the nature of the ground-state within 1-loop RG. Exactly at the VHS many channels (SC/CDW/SDW) are degenerate but moving about from perfect nesting generically helps the SC order. Hence the authors focus their attention on the SC order for the later part of the article.

The article is systematically written, well and adequately referenced and the results are contextualized aptly. The work itself is technical and challenging. It is commendable that the authors managed to use symmetries effectively to group the equations effectively and analyze them. The work certainly deserves publication. However, I would like the authors to address the critique below first before I could recommend the manuscript for publication.

Clarifications and suggestions on the manuscript:

1. Fig. 1: The color of the lines in Fig 1b are not explained. I think they are supposed to correspond to propagators residing in different patches. I suggest that the patches in Fig 1a be color coded. This makes the reader relate to the scattering process directly. I would also suggest labelling the l, m, n and u, v -index in the figure for the cases considered. Once again, this will provide an immediate visual clarity to the reader (something along the lines of Fig 3a, perhaps).
2. The strict definition and meaning of self-similarity is formally only presented in Eq 11 on page 8. Yet the abstract and introduction talk about it multiple times along with the statements asserting the main results. While, for an expert reader the meaning is implied, for a broader readership, it might be beneficial to introduce this idea, at least informally, early on.
3. Page 5, end of section B: It is claimed that the self-similarity is one of the main results of the work. Yet, this just seems to be a statement with no real vision for the reader as to where it could be arising from or how to see this self-similarity. In particular, it would be good to schematically present some of the things from the supplementary section to guide the readers as to why this is arising in the RG equations. This is indeed quite interesting, but one is not in a position to realize if this is just a coincidence or there is something about the structure of the equations that is imposing this. Another aspect that was not clear to me is that if the g/\sqrt{q} scaling is true for all p, q or only for the cases the authors explicitly checked ($q=2, q=3$)?. If it is general, is there a proof? This is really not discussed at all in the main text.
4. Page 5, the discussion on $g^{(l)}$'s: I think a (schematic) figure here will help the readers immensely why the said $g^{(l)}$'s contribute to the said vertices.
5. Fig 2: Needs to be improved. I would suggest the main figures to show, in detail, the flow close to t_c and perhaps introduce insets, if needed, that show the larger scale of t . In the current form, nothing is really clear, especially the competition.
6. Providing an interpretation of the RG flows could be beneficial to the reader. Despite the obvious complexity involved here, the structure of equations is similar to the RG equations for patch models used in the literature (which the authors have also cited). In that context, the readers would benefit from knowing the differences and similarities of between the processes leading to the growth of SC/SDW/CDW vertices. At first glance, it looks like the same processes/effects are at play here as in the case of doped Graphene, for example. Would the authors like to elaborate on this point? In that sense, the mechanism isn't fundamentally new, but the modified electronic structure provides a new "landscape" for the same phenomena to show up. If the authors agree with this, I think it should be phrased so as to clarify this to the audience.
7. Eq 9. The form-factor's label with $0/+/-$ convention drops out of the blue without any real explanation. The main-text should be a self-contained text and shouldn't direct the readers to the SM to understand the notations.
8. The topological nature (non-trivial Chern #) of SC is arising solely from the topology of the VHS points (their location), correct? Can something general be said in this regard?
9. Page 9: "powerful renormalization group" is a subjective term. It's useful and insightful, but is a

crude approximation to the actual problem. Probably fRG/nRG/QMC and other numerical techniques are more "powerful".

Questions:

1. Landau level band width: Are the authors referring to the parabolic band's bandwidth before the formulation of Landau levels? The phrasing seems open to interpretation.
2. On page 3, the authors specify a regime $E_Z < \Delta < W$. While the latter condition is clear, I am wondering why the former is not " $<<$ " and only " $<$ ". It seems that if E_Z is comparable, one would have to account for the Pauli-pair breaking effects more seriously.
3. Although this was done for square lattice, I would think that the topology of VHSs would play a role in the context of the fate of the competition. By topology here I only mean the number and location of the VHS points. As we consider systems on different lattices, I would suspect that these would change. Any comments on what type of scenarios would change the balance from degeneracy of all orders to CDW/SDW/SC? Or do the authors think that this would be a general consequence within 1 loop RG?
4. How do $l=0$ and $l=1$ SC differ from each other, if they do?

Optional suggestions:

1. Abstract: The terms self-similar symmetry and trajectory is ambiguous and can mean different things to different people. And hence its use in the abstract seems less effective. It would have been OK, but these terms aren't adequately introduced/described in the introduction either, unless one gets to page 5. I suggest that the terms self-similar and re-entrant SC be introduced such that the reader gets a clear idea of the title and abstract after reading the introduction.
2. I would also define/identify re-entrant SC as SC in the reconstructed electronic levels in the presence of strong magnetic fields. The smallness of the magnetic length re-introduces periodicity across the sample and re-constructs the electronic levels. In this sense, the strong "orbital effect" of the magnetic field is absorbed in the reconstruction, leaving the weaker effects to be treated perturbatively.
3. The discussion on the structure of Δ at the end on section 2 seemed rushed and not at par with the rest of the text. The biggest issue to multiple citations to the SM broke the flow of the text. I don't have a definite recommendation here, unfortunately. But would be nice if the text is a bit more polished in terms of readability here. It seems like a take home message is lacking around the discussion of the results for Δ for the various cases. In the current form, it just seems like a collection of results.

Reviewer #2:

Remarks to the Author:

The authors study a model of Hofstadter bands in the weak coupling limit using an RG treatment and show that Hofstadter superconductivity can be realized under various conditions. The authors propose potential realization for reentrant Hofstadter superconductivity particularly in Moire systems. The possibility of strong field reentrant superconductivity in Moire systems have recently been proposed in Ref. 5. The main advance of the current work compared to Ref. 5 is that the authors consider a purely repulsive interaction whereas Ref. 5 considered an attractive interaction to stabilize the superconductor. In addition, the current work exploits the bandwidth of the Hofstadter band to employ a weak coupling analysis. This requires starting with a model with already sizable bandwidth and focusing on relatively small values of q (which denotes the denominator of the flux p/q). The analysis is sound but there are several issues that need to be addressed by the authors before I can recommend for publication:

1. One of the surprising results of the analysis is that superconductivity seems to become more competitive compared to other states, e.g. charge density waves, in the presence of finite flux compared to the zero flux case. What is the underlying reason for this? And is this a robust general effect or is it specific to the model and values of the flux studies by the authors?
2. The connection to experimental systems is not made very precise. I understand that the paper aims more for a proof of principle by using a simplified model to illustrate the effect, but the authors should at least clarify the experimental settings where their analysis may be relevant. The weak coupling analysis is likely not applicable for Moire systems with flat bands but may be relevant away from the flat band limit, for example away from the magic angle in twisted bilayer graphene. However, it is unclear whether other constraints, such as the requirement that Zeeman field is smaller than the pairing gap and significantly smaller than the bandwidth can be compatible

with small q which generally corresponds to large magnetic field. Is there a realistic physical system where all the constraints of the analysis are applicable?

3. It is unclear what is the physical significance of the number of van Hove singularities in the Hofstadter bands. For instance, we can artificially define an enlarged unit cell and correspondingly fold the BZ leading to increase in the number of van Hove singularities without changing the physics. Is the percentage of states within some energy window around the van Hove singularity larger in the Hofstadter band as a result of magnetic translation symmetry?

Reviewer #3:

Remarks to the Author:

In the manuscript "Unconventional Self-Similar Hofstadter Superconductivity from Repulsive Interactions", Shaffer, Wang, and Santos study superconducting instabilities in the (Hofstadter) bands of a square-lattice Hubbard model with repulsive interactions in the presence of a commensurate flux – a phenomenon referred to as "Hofstadter Superconductivity". They derive a patch-model and study it, using a renormalization group analysis. They find chiral and nodal superconductivity; their renormalization group exhibits a fixed trajectory and leads to an order parameter which they identify as exhibiting a self-similar property. While their work is clearly connected to an earlier and very interesting work (Phys. Rev. B 104, 184501) of the same authors. However, the current manuscript goes significantly beyond that work, as it details the energetic principles leading to unconventional Hofstadter superconductors.

I think the manuscript is well written and timely. While I believe that it is in principle suitable for an elite journal such as Nature Communications, I first would like the authors to address the following comments:

1) I think the authors should explain better/more accurately what the novelty of their work is compared to the existing literature on pairing in Hofstadter bands. For instance, in the introduction they say that they provide a microscopic mechanism. However, in section (II.B) on the actual model they state that, instead of using a model more appropriate for twisted graphene, they use a square lattice model to establish the essential features. As such I am not so sure they really perform a microscopic study. That's fine but should be stated coherently in the manuscript.

2) In my opinion the manuscript should explain in more detail what the precise definition and physical implications of the self-similarity of the RG trajectory are. Has this only been verified for $q=2$ and 3? Had this been missed in the previous analyses of patch models with fewer VHS or is this property absent in the previous RG flows? Similarly, the reason for calling S in Eq. (11) self-similarity and the consequences should be better explained. These are the qualitatively new aspects, while details of the RG approach (which is very well established and, as far as I can see, standard) are less relevant in my view.

3) Right at the end of page 8, the authors write: "In this case symmetry does not completely fix the form of the gap function, but we find that the simplest form ...". It is true that symmetry does not entirely fix it, but energetics should – something that the authors should be able to address in their theory rather than writing down the "simplest form". Or is this impossible due to the patch theory that is being used?

4) A physical question: breaking which point symmetry would be detrimental to the pairing states they find? Is it two-fold rotation as this symmetry guarantees the degeneracy of VHSs that are paired?

5) Two typos: In the sentence before Eq. (11), there are two "as". On page 5: "there are are q or $q/2$ ".

Response to Reviewers - NCOMMS-22-20832

Reviewer 1

We thank Reviewer 1 for their time, and for the valuable questions and comments, which contributed to improving the manuscript. We are encouraged with the Reviewer's recognition that the work deserves publication. Below we provide a detailed point-by-point response to all the questions and comments raised by the Reviewer.

Clarifications and suggestions on the manuscript:

1. Fig. 1: The color of the lines in Fig 1b are not explained. I think they are supposed to correspond to propagators residing in different patches. I suggest that the patches in Fig 1a be color coded. This makes the reader relate to the scattering process directly. I would also suggest labelling the l, m, n and u, v -index in the figure for the cases considered. Once again, this will provide an immediate visual clarity to the reader (something along the lines of Fig 3a, perhaps).

The colors in Fig. 1b have been explained in the caption, we tried to make this more clear in the new version and color coded the patches in Fig 1a as suggested by the reviewer. The magnetic 'flavor' indices ℓ, m, n are shown in Fig. 1c, while the VHS indices u, v are now illustrated in Fig. 1a.

2. The strict definition and meaning of self-similarity is formally only presented in Eq 11 on page 8. Yet the abstract and introduction talks about it multiple times along with in the statements asserting the main results. While, for an expert reader the meaning is implied, for a broader readership, it might be beneficial to introduce this idea, at least informally, early on.

We added a paragraph in the introduction about self-similarity. We thank the reviewer for this suggestion, which contributes to the improvement of the presentation of the manuscript.

3. Page 5, end of section B: It is claimed that the self-similarity is one of the main results of the work. Yet, this just seems to a statement with no real vision for the reader as to where it could be arising from or how to see this self-similarity. In particular, it would be good to schematically present some of the things from the supplementary section to guide the readers as to why this is arising in the RG equations. This is indeed quite interesting, but one is not in a position to realize if this is just a coincidence or there is something about the structure of the equations that is imposing this. Another aspect that was not clear to me is that if the g/\sqrt{q} scaling is true for all p, q or only for the cases the authors explicitly checked ($q=2, q=3$)?. If it is general, is there a proof? This is really not discussed at all in the main text.

We followed the reviewer's advice and replaced the symbolic expression in Eq. (7) with the full equation from the supplementary material. Hopefully this clarifies that the existence of the self-similar fixed trajectory is not accidental and holds for all q (and that the g/\sqrt{q} scaling follows simply by dimensional analysis and 'flavor' counting). One obtains the result by simply taking $g_{mn}^{(\ell),j} = g_j/\sqrt{q}$, in which case it is clear by inspection that the RG equations for $q = 1$ (no magnetic field) are recovered. This is part of the reason we have grouped the interactions into the five processes according to their VHS indices labeled by $j = 1, 1', 2, 3, 4$, which shows how the structure of the zero flux RG equations is inherited by the RG equations for non-zero flux.

We hope it is also clear, however, that the fixed trajectory is not guaranteed to be reached for arbitrary bare interactions. Indeed, among the cases we considered with Hubbard interaction the self-similar fixed trajectory is only reached in the top and bottom Hofstadter bands for $q = 3$ but not in the middle band, nor for either band for $q = 2$.

4. Page 5, the discussion on $g^{(1)}$'s: I think a (schematic) figure here will help the readers immensely why the said $g^{(1)}$'s contribute to the said vertices.

We added a new figure, Fig. 3, that shows the schematic form of the RG vertex flow equations showing which g_j contribute to which vertex flow.

5. Fig 2: Needs to be improved. I would suggest the main figures to show, in detail, the flow close to t_c and perhaps introduce insets, if needed, that show the larger scale of t . In the current form, nothing is really clear, especially the competition.

Fig. 2 a, b and d required modifications due to several typos in our code, which however did not result in significant changes of our results. We note that Fig. 2 b and d, as well as the inset in c, already showed the flow very close to t_c , but part of the confusion may stem from the fact that we re-scaled the t axis in order to make the tick labels more clear. The competition can be inferred from the fact that the red lines (corresponding to pairing channels) take over the green lines (corresponding to the SDW channels). The main change in Fig. 2 b and d after fixing the typo in our code is the change in the flow of the SDW vertices and the scale at which the competition is resolved.

We tried to make Fig. 2 a more clear by not showing the flow as close to t_c , as our main purpose for showing this figure is to show the sign changing of the g_1 coupling constants, and the general qualitative similarity of the RG flow to that for the $q = 1$ case. For this reason we also kept Fig. 2 c unchanged. In Fig. 2 a the change due to fixing of the typo is in the flow of the $g_{mn}^{(1)j}$ coupling constants (the dashed lines), which however do not contribute to the leading instability and thus did not alter our results.

6. Providing an interpretation of the RG flows could be beneficial to the reader. Despite the obvious complexity involved here, the structure of equations is similar to the RG equations for patch models used in the literature (which the authors have also cited). In that context, the readers would benefit from knowing the differences and similarities of between the processes leading to the growth of SC/SDW/CDW vertices. At first glance, it looks like the same processes/effects are at play here as in the case of doped Graphene, for example. Would the authors like to elaborate on this point? In that sense, the mechanism isn't fundamentally new, but the modified electronic structure provides a new "landscape" for the same phenomena to show up. If the authors agree with this, I think it should be phrased so as to clarify this to the audience.

There are some similarities between our results and the doped graphene RG results in that the number of VHS points can alter the RG flow, but there are important differences. Though the fact that for $q = 2$ SC becomes the leading instability is similar to the doped graphene result, we note that the structure of the equations and the resulting SC phase are quite different. In particular the presence of pairing with finite momenta, promoted by the $g_{mn}^{(1)j}$ couplings, alters the fixed point landscape.

A second difference is that for the $q = 3$ case we actually find that the RG flow is in a sense not affected by the increase in the number of the VHS points when the self-similar fixed trajectory is reached (as happens in the top and bottom Hofstadter bands). In that case our result is opposite from the conclusion of the doped graphene RG calculation. Importantly, though the flow is the same as for $q = 1$, the resulting gap function is quite different and gives rise to chiral SC, as we discuss. The chiral SC is also distinct from the phase found in doped graphene, for which the TRS is broken spontaneously. In our case TRS is explicitly broken with electronic pairing developing in a topological Chern band.

In general, the MTG symmetries make the structure of the RG flow significantly different from some of the previously considered cases. We believe those symmetries are underlying the novel effects that we find and hence these effects have a distinct origin from those found in previous works.

7. Eq 9. The form-factor's label with 0/+/- convention drops out of the blue without any real explanation. The main-text should be a self-contained text and shouldn't direct the readers to the SM to understand the notations.

We agree with the reviewer that the notation labeling the irrep of the gap function is unnecessarily complicated for the main text. We changed it to (*d*) for 0/+/- and (*s*) for 0/+/+.

8. The topological nature (non-trivial Chern #) of SC is arising solely from the topology of the VHS points (their location), correct?
Can something general be said in this regard?

In our framework, it is non-trivial to establish a direct connection between the topology of the VHS and the topological nature of the emergent SC. The reason being that the RG analysis focuses on the dominant contribution arising from a discrete set of VHSs, while the determination of the topological character of the SC requires an analysis of the momentum-dependence of the order parameter around the Fermi surface, which take into consideration point group symmetries and properties of the band structure.

For example, at $q = 2$, we arrived at the d-wave gap function Eq. (9) from a synthesis of (i) the sign of the order parameter calculated from the RG with time-reversal and (ii) the \hat{C}_4 symmetries, leading to our identification of a nodal order parameter. For the self-similar solution (which we verified is realized for $q = 3$), the VHS positions determine the sign changes of the gap function, but since the patch RG calculation does not determine the complex phase of the gap function away from the VHS points, their positions alone cannot explain the chirality. We tackled this question by constructing an order parameter that realizes the self-similar symmetry described in Eq. (11) and subsequent paragraphs. In the projection of the gap from sublattice basis to the band, the wave-functions of the topological Chern band give rise to the complex form factors that carry the phase-winding around the

Fermi surface. However, further analysis is needed to understand if this picture carries over to larger q .

9. Page 9: "powerful renormalization group" is a subjective term. It's useful and insightful, but is a crude approximation to the actual problem. Probably fRG/nRG/QMC and other numerical techniques are more "powerful".

We have slightly rephrased that paragraph (now the second to last paragraph on page 10) to avoid subjective terms.

Questions:

1. Landau level band width: Are the authors referring to the parabolic band's bandwidth before the formulation of Landau levels? The phrasing seems open to interpretation.

We assume this in reference to the first sentence of the second paragraph of the introduction, "In this work we propose that this issue can be circumvented in Hofstadter bands that unlike Landau levels have a finite bandwidth W ...". We agree that the lack of commas separating "unlike Landau levels" makes the phrase open for interpretation, and we added the missing commas to the manuscript. We are saying that Hofstadter bands have a finite bandwidth, unlike Landau levels that have no bandwidth. This is a conceptually important, yet much understated, distinction that our work fully explores to identify a weak coupling regime amenable to RG analysis in Hofstadter bands.

2. On page 3, the authors specify a regime $E_Z \ll W$. While the latter condition is clear, I am wondering why the former is not " \ll " and only " \ll ". It seems that if E_Z is comparable, one would have to account for the

Pauli-pair breaking effects more seriously.

We changed $<$ to \ll to be more conservative. In principle, it is correct that for $E_z \lesssim \Delta$ the Pauli limit has to be considered, but in general we expect this to simply lower T_c of the paired state, with the Pauli limit exceeded only once $E_z \gtrsim \Delta$ (with some numerical factors of order one).

3. Although this was done for square lattice, I would think that the topology of VHSs would play a role in the context of the fate of the competition. By topology here I only mean the number and location of the VHS points. As we consider systems on different lattices, I would suspect that these would change. Any comments on what type of scenarios would change the balance from degeneracy of all orders to CDW/SDW/SC? Or do the authors think that this would be a general consequence within 1 loop RG?

The topology of the VHSs certainly can affect the resulting instabilities. One thing we can state is that since the self-similar trajectory exists for all Hofstadter systems, if that trajectory is reached in the RG flow for systems with other lattices, the resulting instability would be of the same kind as for those lattices. For example, for graphene we expect SC to be the dominant instability already at perfect nesting as found in reference [47], provided the self-similar trajectory is reached. On the other hand, if the self-similar trajectory is not reached, other scenarios are possible. The outcome of course also depends on the type of bare interactions. We plan to investigate these questions in future work.

4. How do $\ell=0$ and $\ell=1$ SC differ from each other, if they do?

For general q , the $\ell = 0$ and 1 Cooper pairs carry different momenta. The $\ell = 1$ SC may thus more accurately be called a pair density wave. For odd $q > 2$, the $\ell = 0, \dots, q - 1$ SC's are all degenerate due to the MTG symmetries (see our earlier work [31]). That is

why for $q = 3$ we had to analyse the fourth order free energy to determine the symmetry of the ground state. For even $q \geq 2$, even and odd ℓ correspond to different irreducible representations of the MTG, and hence form different channels in the one-loop RG. For $q = 2$ in particular, the $\ell = 0$ channel is the usual uniform SC channel, while the $\ell = 1$ is a PDW with ordering momentum $(0, \pi)$.

Optional suggestions:

1. Abstract: The terms self-similar symmetry and trajectory is ambiguous and can mean different things to different people. And hence its use in the abstract seems less effective. It would have been OK, but these terms aren't adequately introduced/described in the introduction either, unless one gets to page 5. I suggest that the terms self-similar and re-entrant SC be introduced such that the reader gets a clear idea of the title and abstract after reading the introduction.

We agree with the reviewer and added a paragraph discussing self-similarity of the Hofstadter system in the introduction (paragraph 4), as well as a qualitative description of the self-similarity properties of the RG flow and pairing gap function that we find.

2. I would also define/identify re-entrant SC as SC in the reconstructed electronic levels in the presence of strong magnetic fields. The smallness of the magnetic length re-introduces periodicity across the sample and re-constructs the electronic levels. In this sense, the strong "orbital effect" of the magnetic field is absorbed in the reconstruction, leaving the weaker effects to be treated perturbatively.

We agree with the reviewer and slightly rephrased the definition of re-entrant SC in the first paragraph of the introduction. We should emphasize that we consider the SC in reconstructed Landau levels (of zero bandwidth) as well as in Hofstadter bands (of finite bandwidth) to be examples of re-entrant SC.

3. The discussion on the structure of Delta at the end on section 2 seemed rushed and not at par with the rest of the text. The biggest issue to multiple citations to the SM broke the flow of the text. I don't have a definite recommendation here, unfortunately. But would be nice if the text is a bit more polished in terms of readability here. It seems like a take home message is lacking around the discussion of the results for Delta for the various cases. In the current form, it just seems like a collection of results.

We hope that by addressing the reviewer's previous comments we have improved that section. Some references to the SM have been removed, but we believe that some of the calculations underlying the results of the section are somewhat non-standard but too long or technical to present in the main text, warranting a reference to the SM. We have also moved the summary of the results in the discussion section to follow immediately the section on the structure of the gap function, which we hope clarifies the take home message.

Reviewer 2

We thank Reviewer 2 for their time, and for the valuable questions and comments, which contributed to improving the manuscript. Below we address the issues raised by the Reviewer.

1. One of the surprising results of the analysis is that superconductivity seems to become more competitive compared to other states, e.g. charge density waves, in the presence of finite flux compared to the zero flux case. What is the underlying reason for this? And is this a robust general effect or is it specific to the model and values of the flux studied by the authors?

Before answering the salient questions raised by the reviewer, we need to clarify our results. While we did find that superconductivity is more competitive than other states for the special case of π flux ($q = 2$), this is not our a general result. Rather, the general result is the identification of a self-similar fixed trajectory of the RG flow for all values of q , in which superconductivity is as competitive as other states in the absence of the magnetic flux. This still makes superconductivity in the presence of the flux more competitive than one naively expects, as the common wisdom is that the magnetic flux is detrimental to superconductivity. More precisely, along the self-similar fixed trajectory the RG flow is the same as the well-studied zero field RG flow of the Hubbard model (studied for example in References 44-46). On the square lattice at Van Hove filling both the SC and SDW vertices diverge in the zero field RG and are degenerate at perfect nesting, with SC winning away from the nesting condition once symmetry allowed terms are included in the model. We stress that, in our opinion, this is a novel, non-trivial result that showcases the power of symmetries (in this case, magnetic translations) in constraining the renormalization flows, despite strong modifications of the band structure due to the presence of the magnetic flux.

Importantly, this self-similar fixed trajectory is only one of several possible fixed trajectories of the renormalization flow. The emergence of a rich landscape of fixed trajectories (signaling a rich phase diagram) is captured by the renormalization of couplings (Eq. 7) due to proliferation of VHS. In particular, the fact that superconductivity wins already at perfect nesting for $q = 2$ is an example when the self-similar trajectory is not reached. We note that, if the microscopic conditions are such that the self-similar trajectory is not realized, one must perform a detailed (numerical) analysis of the RG flows in order to identify the leading instabilities. Remarkably, we have explicitly found that this is indeed realized at $q = 3$ (upon detuning from nesting) with standard Hubbard interactions, leading to a new class of topological superconductor driven only by electronic repulsive interactions which supports chiral Majorana edge states, as shown in Figure 5. To our knowledge, this is the first theoretical analysis to show that topological superconductivity can arise from repulsive interactions in fractal electronic bands.

To further address the generality of our result, we emphasize that the existence of the self-similar trajectory can be generalized to other systems with VHSs with large magnetic fields, since it only relies on the MTG symmetries that are characteristic of Hofstadter systems. We can therefore state that for systems where superconductivity is driven by

the VHS mechanism, superconductivity remains a possible instability (i.e. a fixed point of the RG flow equation) at large magnetic fluxes, at least if Zeeman splitting is small. The latter limitation would likely not apply to systems with triplet superconductivity, which is a subject beyond the scope of this work and that we plan to investigate in a future study. As the generality of our results was not sufficiently clear in the initial manuscript, we edited the text at the end of Sec. II.B to clarify these issues.

2. The connection to experimental systems is not made very precise. I understand that the paper aims more for a proof of principle by using a simplified model to illustrate the effect, but the authors should at least clarify the experimental settings where their analysis may be relevant. The weak coupling analysis is likely not applicable for Moire systems with flat bands but may be relevant away from the flat band limit, for example away from the magic angle in twisted bilayer graphene. However, it is unclear whether other constraints, such as the requirement that Zeeman field is smaller than the pairing gap and significantly smaller than the bandwidth can be compatible with small q which generally corresponds to large magnetic field. Is there a realistic physical system where all the constraints of the analysis are applicable?

As the reviewer rightly marks, the main intent of the work is to establish a proof of principle to a challenging, outstanding question regarding the onset of superconductivity in Hofstadter systems with repulsive interactions. This informed our decision to apply our framework to the square lattice Hofstadter-Hubbard model as archetypal system that captures both orbital and correlation effects, while having a band structure considerably simpler than in typical moiré systems. That being said, this work lays the ground to understand the competition between electronic instabilities in a range of Hofstadter systems. In that regard, we discussed in Sec. III some possible directions to experimental realizations:

1. Optical lattices: In these systems, the bosonic square lattice Hofstadter regimes has been achieved in tilted lattices as reported in Refs. [84-85] that we added in the new manuscript, though experimental progress is needed to achieve larger fermionic systems of cold atoms.

2. Graphene-based moiré systems: These systems have been extensively investigated with most of the work concentrated to the “magic angle” scenario where SC has been observed at zero field, as well as fractal Hofstadter bands at finite fields. We made an estimate for the Zeeman energy scale for MATBG at the end of the discussion, which comes out to less than about 2 meV, based on fields of the order of 10T to produce fluxes in 1/3 and 1/2 regimes. We also note that such small Zeeman splitting has been reported, for instance, by Efetov’s group in Ref. [117]. However, we agree with the reviewer that the weak coupling assumption is less favourable when the Fermi level lies within a flat band, which is the typical target in MATBG. On the other hand, the presence of the magnetic field opens the possibility to explore other regions of the moiré spectrum beyond the usual flat bands. For instance, we note that Ref.[30] has recently made progress in mapping out the spectral properties of Hofstadter moiré systems using scanning single-electron transistor measurements. We find it conceivable that, given the existence of several control knobs in these systems (e.g. magnetic field, twist angle, number/type of layers, electronic gates, etc), one can in future experiments identify Hofstadter bands with finite bandwidth that are favourable to weak coupling scenario discussed in our work. The analysis of this intriguing scenario will require a new derivation of the RG equations taking into account the new landscape of VHS and other details of the band structure and interactions. For example, electronic correlations may actually lead to important band renormalization, as discussed in some recent literature on moiré systems. At any rate, several aspects discovered in this present work will also manifest there, such as the multiplicity of VHS due to the MTG and the presence of self-similar RG trajectories. We had also discussed the extension of our work to the strong coupling regime as a possible future direction, and now added a statement that this would be necessary to properly study Hofstadter superconductivity in magic angle TBG.

3. Twisted bilayer cuprates: One promising possibility we mentioned in the introduction and Sec. III are bilayer twisted cuprates that have recently become possible to produce experimentally. If the upper critical magnetic field H_{c2} in bilayers is comparable to that in the bulk cuprates, it could be around 20 T, high enough to access the Hofstadter regime for realistic twist angles (reference [99] reported monolayer cuprates with T_c similar to the bulk materials, so this may be plausible). We modified the discussion to clarify some of these

points.

3. It is unclear what is the physical significance of the number of van Hove singularities in the Hofstadter bands. For instance, we can artificially define an enlarged unit cell and correspondingly fold the BZ leading to increase in the number of van Hove singularities without changing the physics. Is the percentage of states within some energy window around the van Hove singularity larger in the Hofstadter band as a result of magnetic translation symmetry?

The number of VHSs, and more importantly their positions in the Brillouin zone, is known to profoundly affect the resulting instabilities, doped graphene being a well-known example. In the case of the Hofstadter systems, the increase in the number of VHSs is not the same as the artificial expansion of the Brillouin zone, since in the latter case VHSs in the second Brillouin zone are equivalent to those in the first (i.e. those are the same states). The VHSs in the Hofstadter bands all correspond to distinct states and are located at inequivalent points in the Brillouin zone. The magnetic translation symmetries guarantee that the number of distinct VHSs in Hofstadter systems is always a multiple of q .

Reviewer 3

We thank Reviewer 3 for their time, and for the valuable questions and comments, which contributed to improving the manuscript. We are encouraged by the Reviewer's opinion that this timely work is in principle suitable for an elite journal such as Nature Communications. Below we address the issues raised by the Reviewer.

1) I think the authors should explain better/more accurately what the novelty of their work is compared to the existing literature on pairing in Hofstadter bands. For instance, in the introduction they say that they provide a microscopic mechanism. However, in section (II.B) on the actual model they state that, instead of using a model more appropriate for twisted graphene,

they use a square lattice model to establish the essential features. As such I am not so sure they really perform a microscopic study. That's fine but should be stated coherently in the manuscript.

The reviewer is correct that we do not construct a microscopic model of TBG. By a microscopic model we mean a model in which superconductivity arises from microscopic interactions such as Hubbard interactions, as opposed to mean-field models of superconductivity where the system is simply assumed to be superconducting. This is the main novelty of our work: while previous work simply assumed the existence of pairing in Hofstadter bands and studied its properties if it were to be realized, we show that this can actually happen in a ground state of a simple model with realistic repulsive interactions. To establish this nontrivial result that contradicts conventional expectation that large orbital effects are detrimental to SC, we chose to perform our analysis in a simplified setting that, nevertheless, captures the two main features of the system: orbital and correlation effects. As such, we believe our choice of microscopic model is warranted despite the use of a band structure different from TBG. As we discuss in the introduction and the discussion, our particular model is more suitable to describe systems with square symmetry, like twisted cuprates, but it is a general proof of principle that Hofstadter superconductivity is a plausible ground state. Finally, we believe the framework developed in this first analysis will inform future studies of Hofstadter superconductivity in other systems, including graphene multi-layers.

2) In my opinion the manuscript should explain in more detail what the precise definition and physical implications of the self-similarity of the RG trajectory are. Has this only been verified for $q=2$ and 3 ? Had this been missed in the previous analyses of patch models with fewer VHS or is this property absent in the previous RG flows? Similarly, the reason for calling S in Eq. (11) self-similarity and the consequences should better explained. These are the qualitatively new aspects, while details of the RG approach (which is very well established and, as far as I can see, standard) are less relevant in my view.

The renormalization analysis shows that in the presence of magnetic translation symmetries, the number of independent VHS scales with q , giving rise to a tunable manifold of low energy states. Despite the apparent complexity of the RG equations, by grouping the coupling constants into the $g_{mn}^{(\ell)1}$, $g_{mn}^{(\ell)1'}$, $g_{mn}^{(\ell)2}$, $g_{mn}^{(\ell)3}$, $g_{mn}^{(\ell)4}$ processes according to the VHS patch index structure, we see that the form of these equations is similar to the RG equations in the absence of the magnetic flux, i.e. for $q = 1$. To clarify this point, as well as a similar issue brought up by another reviewer, we replaced the symbolic RG flow equations Eq. (7) with the full expressions including all magnetic flavor indices. The question raised by the reviewer about the precise definition of the self-similar RG trajectory is answered in the manuscript: “fixed point trajectory of Eq. (7) characterized by $g_{mn}^{(\ell)j} = g_j/\sqrt{q}$, i.e. coupling constants independent of the magnetic flavor indices and depending only on the VHS patch indices.” This hopefully clarifies that the existence of the self-similar fixed trajectory holds for all q . This should also clarify that the self-similar fixed trajectory exists as a consequence of the magnetic translation symmetries, which is a characteristic feature of the Hofstadter model that is absent in systems previously studied using the patch RG method. Interestingly, it shows that one (of the many possible) RG trajectories at finite flux “remembers” about the flow at zero field, a fact, in our opinion, that is highly non-trivial. Therefore, provided this fixed trajectory is realized, it provides a one-to-one correspondence between the flow of coupling constants (including the critical exponents near t_c) at zero and finite flux. Although the reviewer is correct that the RG approach is well-established, the self-similar fixed trajectory is a non-standard consequence of the RG flow, which we believe is a novel feature worth emphasizing.

Regarding the physical implications of the self-similarity of the RG trajectory, we added a clarification for calling \hat{S} a self-similarity symmetry right below Eq. (11): under this symmetry, the gap function repeats in momentum space (hence self-similar), but the action in real space is non-trivial and highly non-local. We have shown, for $q = 3$, that the resulting order parameter yields a topological superconductor driven by renormalization of repulsive interactions. These are the main physical consequences of the self-similar trajectory of the RG equations that we identified and which should have a clear experimental signature.

3) Right at the end of page 8, the authors write: \In this case symmetry does

not completely fix the form of the gap function, but we find that the simplest form ...". It is true that symmetry does not entirely fix it, but energetics should { something that the authors should be able to address in their theory rather than writing down the "simplest form". Or is this impossible due to the patch theory that is being used?

The reviewer is correct: in principle the form of the gap function is fixed by the energetics, but the patch RG method only determines the gap function at the VHS patches since states outside of the patches are neglected. This is why it is standard (see e.g. our reference [59]) in the literature to extend the gap function away from the patches based on symmetry considerations. For $q = 2$, the symmetry constraint gives a unique nearest-neighbor form of the gap function, but for $q = 3$ this is not the case and we took the simplest form consistent with the symmetries.

4) A physical question: breaking which point symmetry would be detrimental to the pairing states they find? Is it two-fold rotation as this symmetry guarantees the degeneracy of VHSs that are paired?

The main symmetry that guarantees a logarithmic pairing instability is inversion which guarantees that electron states on the Fermi surface at \mathbf{k} and $-\mathbf{k}$ are degenerate (have the same energy), which is present in the Hofstadter model. We thank the reviewer for bringing this question to our attention; we had earlier referred to inversion symmetry implicitly as one of the point group symmetries but we now mention it explicitly in section II.A for better clarity. As we do state in the text, we are also neglecting the Zeeman term that breaks spin-rotation symmetry (TRS being broken already except for $q = 2$) and would also be detrimental to the spin-singlet pairing states we find.

5) Two typos: In the sentence before Eq. (11), there are two "as".
On page 5: "there are are q or $q/2$ ".

Thank you for pointing the typos out, these have been fixed.

Reviewers' Comments:

Reviewer #1:

Remarks to the Author:

I have read the authors' response to all the referee comments. They did a good job of responding responsibly to all the concerns. While I am happy with the improvements I can't help but notice that (probably as a result of fixing the typo) The instabilities at different fillings have changed. It remains true that SC emerges as winning upon doping (deviation from perfect nesting). But, nevertheless the results for the leading instabilities have changed. And makes a bit more sense in the present form that the SDW orders win.

While I thank the authors for color coding those changes, I would have liked to see them discuss the table and the implications a bit more conspicuously in the response. Given the complexity of the work, it is not easy for me to "check" that. But could the authors clarify the nature of the typo in the code. Was it purely a coding error, or something wrong in the equations that got corrected. I don't think lack of this explanation of change is grounds for dismissal, but it does raise the question: how do we know this is the correct version?

However, I think one aspect still remains unclear/unanswered to me. This is also partially reflected in the reviewer 2's question on the physics behind the competitions. This thing is the following: the authors state that the fact that at $q=2$ SC wins seems accidental, but the self-similar nature of the RG equations is beneficial for SC in general. This is something I think the authors should try to show/argue more concretely. I think this is a strong and important statement in the context of superconductivity as a phenomenon itself. However, I couldn't see what in the self-similar equations is giving a boost to SC. What is the extra channel? Is this just a game of numbers? Could there be some physics behind this? Because this would end up being the main take-away even for a novice reader. Right now, it is more of a "this is what we got" message. Could we understand that statement about boosting SC heuristically somehow?

It will be difficult for anyone else to address this question, but since the authors have worked intensively on this, I think they are in the best position to answer this and hence the push.

Reviewer #3:

Remarks to the Author:

I think the authors have addressed all of my questions and the current version of the manuscript can be accepted for publication.

Response to Reviewers - NCOMMS-22-20832A

Reviewer 1

We thank Reviewer 1 for again carefully reading our manuscript and responses.

I have read the authors' response to all the referee comments. They did a good job of responding responsibly to all the concerns. While I am happy with the improvements I can't help but notice that (probably as a result of fixing the typo) The instabilities at different fillings have changed. It remains true that SC emerges as winning upon doping (deviation from perfect nesting). But, nevertheless the results for the leading instabilities have changed. And makes a bit more sense in the present form that the SDW orders win.

While I thank the authors for color coding those changes, I would have liked to see them discuss the table and the implications a bit more conspicuously in the response. Given the complexity of the work, it is not easy for me to "check" that. But could the authors clarify the nature of the typo in the code. Was it purely a coding error, or something wrong in the equations that got corrected. I don't think lack of this explanation of change is grounds for dismissal, but it does raise the question: how do we know this is the correct version?

We apologize for an oversight on our part, in that we had explained the typos in our code in our response to the editor that the reviewer could not access. The following is our earlier message to the editor, explaining the list of corrections we had made:

In addition, we have reviewed our manuscript and code ourselves and identified several errata. Though none of these qualitatively affect our results, there are several important changes that we had to make. The most important change is due to a typo in Eq. (S7) in the supplementary material in the RG equations for the flow of the SDW vertex. As a result, the leading instability in the middle band for $q = 3$ is in fact an SDW rather than

a CDW. Related to this, there is a smaller typo related to the symmetry properties of the density wave channels under magnetic translation symmetries (a missing factor of $1/2$ in Eq. (S6)). We have also identified a typo in the code for the flow of coupling constants $g_{mn}^{(\ell)}$ for $\ell = 1$, resulting in a slightly different flow in Figure 2(a) but no change to the main results of our work. Fixing these errors required several changes to the main text, including changes in Table I and Figure 2 (a), (b) and (d) (some of the changes to the figure were also in response to suggestions by Reviewer 1). There was also a small typo in Eq. (10). We emphasize that these corrections do not change our main conclusions.

To address the reviewer's question of how we know that the corrected version is indeed correct, we tracked down where the error occurred due to a copy/paste error in the code. As a check, we note the corrected RG flow also correctly reproduces the degeneracy of SDW channels due to hermiticity (see discussion under Eq. (8) in main text), and as the reviewer notes it makes more sense that SDW wins rather than a CDW. We apologize again for not addressing the errors in our original response to the reviewers.

However, I think one aspect still remains unclear/unanswered to me. This is also partially reflected in the reviewer 2's question on the physics behind the competitions. This thing is the following: the authors state that the fact that at $q=2$ SC wins seems accidental, but the self-similar nature of the RG equations is beneficial for SC in general. This is something I think the authors should try to show/argue more concretely. I think this is a strong and important statement in the context of superconductivity as a phenomenon itself. However, I couldn't see what in the self-similar equations is giving a boost to SC. What is the extra channel? Is this just a game of numbers? Could there be some physics behind this? Because this would end up being the main take-away even for a novice reader. Right now, it is more of a "this is what we got" message. Could we understand that statement about boosting SC heuristically somehow?

It will be difficult for anyone else to address this question, but since the authors have worked intensively on this, I think they are in the best position to answer this and hence the push.

We appreciate the reviewer’s push for us to improve our work and clarify the presentation. First, it appears that there may have been a bit of a misunderstanding concerning the $q = 2$ case and its relation to the self-similar RG fixed trajectory. To clarify, the self-similar fixed trajectory is in fact not reached for $q = 2$, indeed providing an example of when SC decisively wins even with perfect nesting in the SDW channel which does not happen along the self-similar fixed trajectory (where at perfect nesting SC and SDW are degenerate). Though we did not previously state that the self-similar fixed trajectory is not reached for $q = 2$, we never stated that it was. We understand that this may have been unclear, so we added explicit statements that the self-similar fixed trajectory is not reached for $q = 2$ (at the end of section II.B, in section II.D on page 7, and in the first paragraph of section III). Nevertheless, we would not say that the fact that SC wins at $q = 2$ is accidental, rather this SC instability corresponds to another fixed trajectory of the RG flow that is realized due to the microscopic properties of the system such as particle filling, band structure and interactions.

To address the second part of the question, we would also not say that the self-similar fixed trajectory boosts superconductivity relative to the case with no magnetic field. Rather, the self-similar fixed trajectory (itself a consequence of the self-similar nature of the Hofstadter band structure due to magnetic translation symmetries) is beneficial to SC in the sense that it guarantees the existence of a SC instability within the microscopic parameter space in the presence of a magnetic flux, provided that instability exists in the absence of a flux. Indeed, along the self-similar fixed trajectory the SC instability is only realized away from perfect nesting in the particle-hole SDW channel, as it happens at zero field.

To reiterate this again, whether or not the self-similar fixed trajectory is reached depends on relevant microscopic considerations and there is in general a large number of fixed trajectories in the RG flow in Eq. (7). This is in fact demonstrated by the $q = 2$ case and the middle band in the $q = 3$ case, for which the self-similar fixed trajectory is not realized. This is not *a priori* good or bad for SC as these cases also demonstrate that either a SC or a SDW instability respectively may be realized away from the self-similar fixed trajectory, resulting in a potentially rich and complex phase diagram. This is indeed why the fact that the self-similar fixed trajectory is in fact reached in the top and bottom bands for $q = 3$ is a very non-trivial result.

Reviewers' Comments:

Reviewer #1:

Remarks to the Author:

I am content with the response from the authors.